



# Petrogenesis and geodynamic implications of Ediacaran rocks from the Sirwa massif (Central Anti-Atlas); insights from U-Pb geochronology, whole-rock geochemistry, and Sm-Nd isotopes

Abdelhay Ben-Tami[1,2], Said Belkacim[1,6], Jamal El Kabouri[1], Bouchra Baidada[3], Joshua H.F.L. Davies[4],
Morgann G. Perrot[5], Mohamed Bhilisse[2], Mohamed Assalmi[2], Mariam Ferraq[1,6], Mohamed
Bouabdellah[7,8], David Lalonde[2].

[1] Laboratoire de Géologie Appliquée et Géo-Environnement, Département de la Géologie, Faculté des Sciences, Université Ibn Zohr, B.P.8106, Agadir, Maroc.

[2] Zgounder Millennium Silver Mining, Aya Gold and Silver Group, Rue De L'epargne, Quartier Racine, Casablanca, Maroc.

[3] Higher school of Technology, Fkih Ben Salah, Beni Mellal, Morocco.

[4] Department of Earth and Atmosphere Sciences, University of Quebec in Montreal, Montreal, QC H3C 3P8, Canada.

[5] Department of Earth and Planetary Sciences, McGill University, Montreal, Quebec H3A 0E8, Canada.

[6] Mining and Environmental Research Institute, University of Quebec in Abitibi-Timiskaming, Rouyn-Noranda, QC J9X 5E4, Canada.

[7] Laboratoire des Gîtes Minéraux, Hydrogéologie et Environnement, Faculté des Sciences, Université Mohammed Premier, 60000, Oujda, Maroc.

[8] Geology and Sustainable Mining Institute, Mohammed VI Polytechnic University, Benguerir 43150, Morocco.

*Correspondence to*: Abdelhay Ben-Tami (abdelhaybentami@gmail.com)

## Abstract.

The geodynamic evolution of the Anti-Atlas belt post-Pan-African orogeny (~650 Ma) remains debated, particularly regarding the basement beneath the Central Anti-Atlas, and the geological processes leading to the formation of the Ediacaran Saghro Group (SG), and Ouarzazate Group (OG). New LA-ICP-MS U-Pb ages of 575 ± 3 Ma and 564 ± 2 Ma were obtained respectively from samples Zg-106, and Zg-119 from the OG. In addition, detrital zircons from SG sediments yield a prominent 2.1 Ga age peak, indicating local recycling of Paleoproterozoic basement material. Geochemically, two magmatic series are

identified : (i) a SG mafic-intermediate calc-alkaline series with Nb-Ta and Ti negative anomalies from early back-arc basin setting; and (ii) a felsic-intermediate high-K calc-alkaline to shoshonitic series of the OG, exhibiting continental magmatic arc signatures. Isotope data (εNd (t): +3.2 to +4.5, TDM = 1431 - 1197 Ma for SG; εNd (t): -0.9 to +1.1, TDM = 1526– 1252 Ma for OG), indicates that the SG formed from a dominantly juvenile, mantle-derived source, with limited crustal contribution; while the slightly younger OG involved significant reworking of older, evolved continental crustal material.

These findings sustain a model where Early Ediacaran SG sediments and associated mafic-intermediate volcanics were formed in a back-arc basin. During this basin development, its shoulders were locally formed by the 2.1 Ga Paleoproterozoic basement, supplying Paleoproterozoic zircons to the Saghro host basin. This, further supports the occurrence of the Eburnian basement north of the Anti-Atlas Major Fault (AAMF). Additionally, the younger OG reflects a Late Ediacaran continental crust collapse event involving widespread crustal reworking and the emplacement of a Silicic Large Igneous Province (SLIP).



**Keywords**

Saghro Group; Ouarzazate Group; Sirwa massif; LA-ICP-MS U-Pb on zircon; 2.1 Ga Paleoproterozoic crust

## 1 Introduction

The Anti-Atlas belt of Morocco in the northern margin of the West African Craton (WAC) bears witness to a long-lived geological evolution started since the break-up of Rodinia. The Tonian - Cryogenian evolution of this belt record the establishment of a passive margin, and island arc which were accreted to the WAC margin at ca. 650 Ma, with the emplacement of syn-tectonic granitoids (Linnemann et al., 2019, 2014; Pereira et al., 2012a; Abati et al., 2010; Liégeois et al., 2006; Gasquet et al., 2005). The subsequent evolution of the Anti-Atlas belt during the Ediacaran is still debated despite the improvements in the understanding of the multiple events that prevailed during Neoproterozoic times (D'Lemos et al. 2006; Samson et al. 2004; Gasquet et al. 2008, 2005, 2004; Thomas et al. 2004, 2002; Ennih and Liégeois, 2008, 2001; Walsh et al. 2002), and led to the final amalgamation of Gondwana (D'Lemos et al., 2006; Gasquet et al., 2008, 2005). Within this time frame, and based on paleogeographic reconstruction and lithostratigraphic correlations; the Anti-Atlas belt was close to the Cadomian subduction system. In liaison, this subduction is responsible for the accretion of peri-Gondwana terranes and the final amalgamation of the Gondwana (El Kabouri et al., 2025; Rojo-Pérez et al., 2024; Stern, 2024; Garfunkel, 2015; Hefferan et al., 2014). However, it is still debated whether the Anti-Atlas records the post-collisional phase of the Pan-African belt, or if it is related to the establishment of a back-arc basin induced by the southward-dipping Cadomian subduction beneath the amalgamated WAC-Iriri/Tazigzaout arc complex at ca. 620 Ma (Errami et al., 2021a; Walsh et al., 2012; El Hadi et al., 2010; Abati et al., 2010). Furthermore, the nature of the basement north of the Anti-Atlas Pan-African suture zone, known as the Anti-Atlas Major Fault is also still a matter of debate. However, in the Zgounder Mine Region, a well-preserved section of Ediacaran sedimentation and magmatism provides key insights into the final stages of Pan-African tectono-magmatic evolution along the northern margin of the WAC.

The study area, so-called the "Zgounder Mine Region" in this contribution corresponds to the vicinity of the Zgounder Ag-Hg deposit (Ben-Tami et al., 2024). The Zgounder deposit is situated at approximately 265 km east of Agadir, and 220 km west of Ouarzazate, along the southern flank of the Ouzellagh-Sirwa Salient in the Central Anti-Atlas Mountains of the WAC. The Ediacaran period of the Zgounder Mine Region (630-539 Ma), is represented by : (i) the Saghro Group (SG) sedimentary successions and coeval mafic to intermediate units deposited at around 630 to 600 Ma (Abati et al., 2010); (ii) large felsic plutons, subvolcanic formations, along with coeval pyroclastic rocks referred to as the Ouarzazate Group (600 Ma - 539 Ma) (Thomas et al., 2002).

We present new whole-rock geochemistry, and Sm-Nd isotopes for the Ediacaran successions of the Zgounder Mine Region. The aims are to decipher their petrogenesis, explore magma sources, and investigate their geodynamic significance in Ediacaran evolution of the WAC. We also report new LA-ICP-MS U-Pb dating on magmatic and detrital zircons to constrain





the lower Ediacaran magmatism and challenge the source of the Saghro Group's sedimentary units, hence, refining existing geodynamic models.

## 2 Geological setting

### 2.1 The Anti-Atlas belt

The Anti-Atlas Mountains, located on the northern edge of the WAC, and consist of a Proterozoic basement and a Late Ediacaran to Paleozoic cover (Leblanc and Lancelot, 1980) (Fig. 1B). These mountains record two major orogenic cycles: the Paleoproterozoic Eburnean and the Neoproterozoic Pan-African orogenies.

The Paleoproterozoic Eburnean orogeny is preserved in the Western Anti-Atlas, where low - to high-grade metamorphic rocks are intruded by a series of granitoids dated at ~2.2 Ga (Hefferan et al., 2014, and references therein; O'Connor et al., 2010; El

Hadi et al., 2010; Ennih and Liégeois, 2008; Gasquet et al., 2008, 2005; Thomas et al., 2002; Walsh et al., 2002; Aït Malek et al., 1998). Overlying these terranes, is the carbonate and quartzite succession of the Taghdout Group deposited in a passive margin (Errami et al., 2021a; Álvaro et al., 2014; Abati et al., 2010; Thomas et al., 2004). U-Pb ages on a quartzite sample from the Taghdout Group provided ages spanning from 2182 to 1987 Ma (Walsh et al., 2012). Moreover, undeformed Doleritic dikes intrude the Taghdout Group in the Ighrem and Zenaga inliers, and were dated between 1710 and 1630 Ma by Ikenne et

al., (2017), and Ait Lahna et al., (2020), respectively. Overall, this implies that the Taghdout Group is in fact of Paleoproterozoic in age (Ikenne et al., 2017; Abati et al., 2010).

The Neoproterozoic episode known as the Pan-African orogeny is well established in the Central and Eastern Anti-Atlas (ca. 760 to 550 Ma) in numerous studies (Bouougri et al., 2020; Soulaimani et al., 2018; Triantafyllou et al., 2016; Karaoui et al., 2015; Blein et al., 2014; Álvaro et al., 2014; Hefferan et al., 2014; Walsh et al., 2012; El Hadi et al., 2010; Errami et al., 2009;

Michard et al., 2008; Gasquet et al., 2008, 2005; Ennih and Liégeois 2008, 2001; Thomas et al., 2004, 2002). These Pan-African successions start with an early Tonian - Cryogenian syn-rift units, consisting of carbonates and quartzites known as the Jbel Lkst Group (Kerdous inlier), the Tachdamt Group (Zenaga inlier), and the Bleida quartzites (Bou-Azzer-El Graara inlier) (Álvaro et al., 2014). The rifting process is evidenced by the formation of oceanic basement now preserved as ophiolitic sequences in the Bou-Azzer and Sirwa inliers (Thomas et al., 2002, 2004). Concurrently, a long-lived island arc complex

formed north of the WAC (Admou et al., 2012; D'Lemos et al., 2006; Thomas et al., 2002, 2004). This arc is now preserved as the 743 ± 14 Ma Iriri Arc in the Sirwa inlier (Thomas et al., 2002), and its equivalent, the Tazigzaout-Bougmane Complex in the Bou-Azzer inlier, dated at 752 ± $^1_2$ Ma (D'Lemos et al., 2006). Following this, and as the north dipping subduction ceased, these latter terranes were subsequently deformed and obducted to the northern margin of the WAC around 650 Ma during the main Pan-African orogeny (Hefferan et al., 2014, and references therein; Thomas et al., 2002). Further,

widespread calc-alkaline magmatic intrusions and regional metamorphism affected the Anti-Atlas belt (Hefferan et al., 2014, and references therein), with several syn-orogenic granitoids dated between 680 and 640 Ma (Hefferan et al., 2012; Walsh et al., 2012; El Hadi et al., 2010; Inglis et al., 2005; Thomas et al., 2002).



In the northeastern domain of the Anti-Atlas, the Ediacaran stratigraphy (630 – 539 Ma) comprises an early Ediacaran basement of Saghro Group, consisting of folded meta-sedimentary units under greenschist facies conditions, and covered by a late-

Ediacaran Ouarzazate Group sequence of volcanic and volcano-sedimentary nature (Ouabid and Garrido, 2023; Errami et al., 2021a; Yajioui et al., 2020; Michard et al., 2017; Blein et al., 2014; Álvaro et al., 2014, 2014b; Walsh et al., 2012; Abati et al., 2010; Errami et al., 2009; Gasquet et al., 2008; Liégeois et al., 2006; Thomas et al., 2002; Fekkak et al., 2001; Bajja, 2001; Ouguir et al., 1996). The transition from volcanic dominated successions of the Ouarzazate Group to the establishment of a stable platform series is recorded during the Ediacaran - Cambrian transition, with the deposition of the Taroudante and Tata

groups in an anorogenic setting (Álvaro et al., 2014).









**Figure 1: (A) Sketch map locating the Anti-Atlas belt in respect to Morocco (Saadi, 1985). (B): geological map of the Central and Eastern Anti-Atlas Proterozoic inliers (redrawn and modified after Michard et al., (2017)). (C): Schematic representation of the geology of the Sirwa massif (compiled and redrawn after Thomas et al., (2002)); location of the Zgounder Mine Region in red square (see Fig. 2). Abbreviations: Taz : Tazoult quartz-porphyry (559 ± 6 Ma); Im: Imourkhssen granite (562 ± 5 Ma); Ti: Tikhfist rhyolite (571 ± 8 Ma); Ask: Askaoun granodiorite (575 ± 8 Ma); Ig: Ighrem granite; Og: Ougougane granite; and SAF: South Atlas Fault. U-Pb ages are from Thomas et al., (2002).**

## 2.2 Sirwa inlier

The recent geological mapping of 1/50 000 scale sheet maps of Douar Çour, Assarrag, Tachoukacht, Tamallakout, Sirwa, Taghdout, and Açdif (see Fig. 1C for relevant sheets) has subdivided the Sirwa inlier into various groups (Thomas et al., 2000a; De Beer et al., 2000; Gresse et al., 2000).

Paleoproterozoic rocks are the oldest in the Sirwa inlier, mainly composed of altered iron-rich gneiss and schist fragments from the Zenaga Complex (Thomas et al., 2002). The Tachdamt Group's sedimentary and volcanic rocks were deposited after Rodinia's breakup (Thomas et al., 2002). Interbedded Tachdamt Group volcaniclastic deposits date to ca. 883 Ma (Bouougri et al., 2020). This Group is overlain by the Khzama Complex, which includes the Tasriwine ophiolite, the Iriri Migmatite, and Tachoukacht schists formed during island arc development and associated back-arc oceanic crust (Thomas et al., 2002).

The Saghro Group, as defined by Thomas et al., (2002), is a thick, ~8000 m pile of deformed sedimentary, volcanoclastic, and volcanic rocks with a calc-alkaline composition and greenschist-facies metamorphism. It includes six lithostratigraphic formations: Tittalt, Agchtim, Tizoula, Imghi, Azarwas, and Tafiat. The lower sequence features greywacke/turbidites with volcanic rocks, while the upper formations consist of coarse-grained clastic rocks (Thomas et al., 2002). The age of the Saghro Group has been constrained by Liégeois et al., (2006) and Abati et al., (2010) using U-Pb dating on detrital zircons from both the Sirwa and Saghro massifs. These studies established a maximum depositional age at approximately 610 Ma. More recent data from the Eastern Saghro massif reported by Errami et al., (2021a), provided slightly younger detrital ages clustering around 607 ± 6 Ma and 604 ± 5 Ma. In addition, the entire Saghro series was intruded by syn-tectonic granitoids dated at 603 ± 6 Ma and 600 ± 3 Ma (Errami et al., 2021a). Crucially, in the Sirwa inlier, the Saghro Group is significantly older as it is intruded by the Mzil granite dated at 614 Ma (Thomas et al., 2002, 2004).

The Bou Salda Group is a thick volcano-sedimentary succession from 610 to 580 Ma (Belkacim et al, 2017; Gasquet et al., 2008), interpreted as basin infills of grabens and pull-apart basins within the AAMF (Errami et al., 2021a; Gasquet et al., 2008; Thomas et al., 2002). In the Sirwa inlier, it mainly comprises the Lmakhzene and Ighil members (Thomas et al., 2002). The Lmakhzene Member features arkosic gritstones, sandstones, and conglomerates, while the volcanic Ighil Member includes basalt, rhyolite, and andesite. Ages for the Bou Salda Group in the Sirwa inlier are derived from the Tadmant and Tamriwine rhyolites, dated respectively at 606 ± 6 Ma and 605 ± 9 Ma (Thomas et al., 2002) (Fig. 1C, and Fig. 2). Pelleter et al. (2016), also dated the Tadmant rhyolite at 610 ± 7 Ma (Fig. 2). Nonetheless, Abati et al., (2010) reported detrital zircon ages of 600 ± 12, 603 ± 13, and 625 ± 12 Ma, suggesting a maximum age for quartzite clasts in a conglomerate and, hence for the Bou Salda Group.





The Ouarzazate Group sequences (580-539 Ma; Blein et al., 2014b; Toummite et al., 2013; Thomas et al., 2004) of immature, coarse clastic sedimentary rocks (conglomerates, arkoses, reworked volcanic rocks) acid to intermediate volcaniclastic rocks (lapilli tuff, volcanic breccias, ignimbrites, etc.) and lavas (minor basalt, andesite and rhyolite) are associated with post-collisional high-K calc-alkaline to shoshonitic magmatism (Soulaimani et al., 2018). Briefly, throughout the Sirwa inlier, the

Ouarzazate Group is subdivided into four subgroups (Fig. 1C for relevant groups) (Tiouin, Bouljama, Tafrant and Achkoukchi) (Thomas et al., 2000a; Gresse et al., 2000; De Beer et al., 2000). Under an extensional regime, typical foreland basin successions of Tata Group were deposited, following the post-orogenic molasse volcaniclastic rocks of the Ouarzazate Group (Thomas et al., 2002) (Fig. 1C).







Figure 2: Detailed geological map of the Zgounder Mine Region (this study). Locations of surface samples are indicated on the map. Underground samples (e.g. Mine levels, drill holes) including the dated Zg-119 are not represented on the map.



## 3 Data and methodology

### 3.1 U-Pb zircon geochronology

Three samples from the Zgounder Mine Region (Zg-106; Zg-119; Zg-132) were selected for U-Pb zircon geochronology (see
Fig. 2 for location: Zg-106, and Zg-132). They were crushed, sieved to 50-250 μm, and weighed (1.4 kg for Zg-106; 1.5 kg for Zg-119; 2.3 kg for Zg-132), at the preparation unit of the Département de Géologie, Faculté des Sciences, Université Ibn Zohr, Maroc. Samples were then shipped to the Geotop Research Center, Université de Québec à Montréal (UQAM), Canada. Analytical parameters are detailed in supplementary data 1, following Horstwood et al. (2016) (refer to table 1 in supplementary data 1). Ages of intrusive rocks are reported as weighted mean $^{206}Pb/^{238}U$ dates, and detrital zircon data as
concordia dates. LA-ICP-MS U-Pb data are in supplementary table 5.

### 3.2 Whole-rock geochemistry and Sm-Nd isotopes

Fresh rock samples were collected for whole-rock geochemistry and Sm-Nd isotopes from outcrops (surface and underground), as well in various diamond drill holes from the Zgounder Mine Region. Crushing and powdering of the rock samples were performed at the facilities of the Zgounder Millennium Silver Mining Company (ZMSM). For each sample, lithostratigraphic
position, sample type, drill core name and depth, short description and analytical methods applied are reported in supplementary table 3. Further details on the analytical procedures are given as supplementary data 2.

## 4 Results

### 4.1 Petrography

#### 4.1.1 The Saghro Group rocks

▪ The Saghro Group is mainly represented by fine to medium-grained, and dark to olive green mafic volcanic rocks. They contain fine-grained plagioclase and clinopyroxene phenocrysts dispersed in the groundmass (Fig. 3). A medium- to coarse-grained equigranular and dark green gabbroic facies is also present, and shows heavily altered pyroxenes and amphiboles (Fig. 3a). Dolerite sample (Fig. 3b) have an intergranular texture, with lath shaped plagioclase, primary Fe-Ti oxides, and clinopyroxene aggregates. Sample Zg-108 is andesitic in composition (Fig.
3c), and shows a medium-grained intergranular texture. The texture is microlitic to microlitic-porphyritic for basalt and basaltic-andesite samples (Fig. 3d).

▪ The dated sandstone sample (Zg-132), displays distinct well-defined, millimeter-scale laminations, controlled by compositional variations. Darker bands are enriched in chlorite, whereas lighter laminae are mainly composed of quartz and muscovite (Fig. 3e). Such alternating patterns suggest fluctuating depositional conditions, specifically
changes in hydrodynamic energy and the amount of clay input, consistent with a low-energy, suspension-dominated sedimentation typical of distal turbidites or quiet shallow-marine environment.



### 4.1.2 The Ouarzazate Group rocks

- Rhyolite samples show a microcrystalline locally glassy texture. Primary mineral assemblages are composed of quartz megacrysts and, K-feldspar altered to albite, and embedded in a devitrified matrix. Secondary alteration minerals are sericite, albite, with some local carbonate veinlets (Fig. 3f). Dolerite samples (Zg-13, Zg-15; Fig. 3g) are fine-grained, exhibiting an intergranular texture. Plagioclase crystals, locally transformed to albite are moderately sericitized and rarely altered to epidote, with equant opaque grains enclosed in the primary minerals (plagioclase, pyroxene). A dolerite sample (Zg-14) collected next to the mineralized zone have an intergranular texture in which laths of black oxidized plagioclase were included in pyroxene crystals (Fig. 3h). The plagioclase crystals are strongly altered and replaced by a mixture of blue chlorite, weak and sparse grains of epidote. Clinopyroxene is mainly replaced by hornblende and actinolite, and partially by pyrite. Ignimbrite samples show an eutaxitic texture with remarkable fiamme and dense welded volcanic glass, along with cryptocrystalline facies locally fluidal, mainly composed of quartz, opaque minerals, sericitized phenocrysts of plagioclase and K-feldspars (Fig. 3i).

- Granodiorite and quartz-diorite samples are grey to light brown with a moderate pinkish tint related to K-feldspar alteration, and they show a homogenous coarse-grained texture. Sericite and clay minerals replace lath-shaped plagioclase phenocrysts, and are associated with chlorite and iron oxides (Fig. 3g). Chlorite replaces biotite, which locally substitutes hornblende crystals. Opaque minerals are enclosed in hornblende, biotite and plagioclase. Granite is medium- to locally fine-grained; it is distinguished from the granodiorite and quartz-diorite by its white to pink color (Fig. 3k).





| Lithology | Sample code | Texture | Primary minerals and accessory phases (size + %) | Alteration minerals |
|---|---|---|---|---|
| **Saghro Group rocks** | | | | |
| Gabbro | Zg-103 Zg-105 | medium to coarse-grained/ locally equigranular | Olivine (<0,1mm; <2%); Pyroxene (up to 0,25mm; less than 5%); hornblende;(5-13%); Ca-rich plagioclase (20-35%); Fe-Ti oxides (up to 9%); accessory sulfides (pyrite); <2% | Secondary Amphibole; Green Chlorite as patches (10%); Sericite (up to 15%); epidote (2%) |
| Dolerite | Zg-107 | intergranular | Clinopyroxene, up to 0,5mm; up to 15%); Plagioclase (20-35%); primary Fe-Ti oxides (up to 12%). | Chlorite; minor epidote; rare albite; minor sericite (green schist alteration package)/ secondary amphibole, |
| Andesite | Zg-108 | Medium-grained intergranular | Primary Fe-Ti oxides (up to 10%); Pyroxene (up to 0,2mm; less than 5%); Plagioclase (up to 35%) | Minor sericite, oxides. |
| Basaltic-andesite | Zg-002 | Microlitic-porphyritic | Clinopyroxenes as fine aggregates (0,1mm; up to 10%); Automorphic plagioclase as micro and phenocrysts locally zoned (0,4 - up 1,4 mm; up to 40%); Opaque minerals (<5%) | Chlorite (micro-patches); Sericite (13%); Argillic alteration, minor kaolinite in plagioclase, oxides altering plagioclase. |
| | Zg-003 Zg-004 | | Micro and local phenocrysts of Pyroxene (0,1 – to up to 1 mm; up to 13%); Plagioclase as micro and phenocrysts locally zoned (0,4 - up to 1,4 mm; 35%); oxides altering plagioclase phenocrysts; rare apatite | Chlorite; dispersed calcite (up to 5%), oxide orioles surrounding pyroxenes; Sericite (14%); |
| Basalt | Zg-006 | Microlitic-porphyritic | Pyroxene (0,1 - 0,8 mm; less than 10%); Micro plagioclase in the mesostasis + phenocrysts (0,2-1,5mm; 30-38%); Fe-Ti oxides. | Abundant calcite, secondary quartz (amygdales); Sericite. |
| Sandstone | Zg-132 | Fine to medium-grained | Compositional variation for darker/lighter laminations: Quartz (0,2-0,4mm; up to 70%); muscovite flakes (10 to 30 %); chlorite (10 to 25 % or more); feldspar (up to 0,3mm); iron oxides (7%). | |
| **Ouarzazate Group rocks** | | | | |
| Rhyolite | Zg-04 Zg-06 Zg-117 | Cryptocrystalline, locally glassy with visible fiamme | Quartz (0, 2-0,4mm; up to 30%); Plagioclase (0, 15-0,6mm; 20-30%); K-feldspar (0, 15-0,8mm; up to 55%); Biotite (~0,1mm; less than 4%); Primary opaque minerals; Zircon (<1%) + xenoliths (1-2mm; 6%) | Slight albitization; Chlorite/muscovite (micro-patches); Sericite; |
| Rhyolitic-ignimbrite | Zg-05 Zg-111 | vitroclastic to cryptocrystalline texture, locally fluidal | Quartz (0,1-1,8 mm; up to 30%); Plagioclase (up to 2,2mm); K-feldspar (0,8-1,9mm); Fe-Ti oxides (less than 4%); + xenoliths (meta-sedimentary facies) | Sericite (4-8%) |
| Porphyritic rhyolite | Zg-110 | porphyritic | Quartz in phenocrysts (1-4,5mm; 18-28%); K-feldspar (up to 4,5 mm); Plagioclase (up to 4,3 mm); Opaque minerals (Less than 5%); + visible xenoliths (up to 2,6mm) | Sericite (3-6%); Muscovite (1-2%) |
| Quartz-diorite | Zg-07 | medium to coarse-grained | Amphibole (0, 15-1mm; 2-8%) ; Biotite (0,25-2mm ; 4-9%) ; Plagioclase (1,3 – 3.4 mm; 32-60%); K-feldspar (up to 1,6m ; up to 8%) ; | Chlorite (5-8%); Muscovite (3%); Epidote (<2%) |



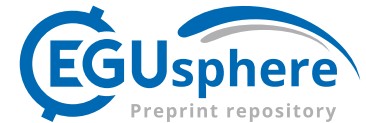

| | | | interstitial quartz (0.2 -1.2 mm; up to 25 %); Primary opaque minerals (inclusion in amphibole crystals; <6%); Zircon (<1%) | Sericite (10-15%); Kaolinite and Clay minerals (2%); Oxides (6%); |
|---|---|---|---|---|
| Granodiorite | Zg-08 | medium to coarse-grained, isogranular texture | Hornblende (0,15-1,2mm ; 6-10%) ; minor Biotite (0,25-2mm ; 4-18%) ; Plagioclase phenocrysts (0.25 – 2.6 mm; 45-55%); Orthoclase (1-2,2m ; 20-35%) ; Quartz (0.4 -2.5 mm; 10-30 %); +xenoliths ; Primary + secondary opaque minerals ; Zircon in biotite (<1%) | Chlorite (less than 10%); Sericite (up to 25%); Epidote (2%);; Clay minerals (2%); Kaolinite (<2%); Oxides (4%) |
| | Zg-09 | | | |
| Dolerite | Zg-13 | intergranular | Clinopyroxene (augite) (0,1mm; up to 6%); brown hornblende (up to 14%); automorphic plagioclase (0,4 -  less than 1,4 mm; up to 43%); Biotite (<4%); secondary Quartz (0,1-0,4mm; <4%); Primary opaque minerals (<5%); accessory sulfides (pyrite); <2% | Intense Chlorite (patches); Sericite (20%); Muscovite (up to 2%) |
| | Zg-15 | | | |
| Diabase | Zg-14 | intergranular | Pyroxene  (0,03-0,1mm; less than 10%); Plagioclase (0,25-2,5mm; 35-40%); Fe-Ti oxides (magnetite+ilmenite prisms, <6%) | Patches of blue chlorite (0,25-1,1 mm; up to 38%); sericite (up to 8%); clay minerals (<6%); Secondary amphibole |
| Ignimbrite | Zg-97 | eutaxitic texture with visible fiamme | Quartz (less than 0,7mm; 45-60%); Plagioclase; K-feldspar (sanidine; up to 23%, 0,4mm); Opaque minerals (euhedral to subhedral) (5-10%); + Xenoliths (anhedral); Zircon in inclusion in k-feldspar crystals (<1%) | Sericite (less than 5%); Scarce chlorite (2%); Kaolinite (5%) |
| | Zg-98 | | | |
| Granophyre | Zg-99 | coarse grained | Amphibole; sub-automorphic Biotite (0,4-0,9mm) ; Plagioclase (up to 8%; 0,5–1mm) ; K-feldspar (microcline) ; Quartz (rounded and eye-shaped (0,4-3,2mm)); Magnetite + apatite+ Zircon (<2%) ( ; +xenoliths (mafic) | Chlorite; Sericite ; Muscovite |
| Rhyolite | Zg-106 | Microcrystalline, locally fluidal | Quartz (0,2-0,4mm) ; K-feldspar (up to 0,3mm); Plagioclase (< 0,4mm); + rare xenoliths (mafic to intermediate). Mesostasis forms more than 85% of the rock. | Albite; Chlorite (3-5%); Sericite (up to 25%); Calcite (veinlets) (2%) |
| Granite | Zg-115 | medium to fine-grained, equigranular | Amphibole (rare 2% (<0,2mm); Biotite (up to 2%); Microcline with visible twinning (20-30% (0,5 to 1,2 mm)); Plagioclase (6-10% (=0,7mm)); Quartz (35-45% (0,4 to 1,3 mm)) ; +metasedimentary xenoliths (0,5-1cm; 3%) ; Opaque minerals; Zircon  (<2%); | Chlorite (moderate); Actinolite (less amount); Sericite (15%); Muscovite (less than 4%) |
| | Zg-119 | | | |
| Rhyolite | Zg-01 | microcrystalline porphyritic texture, | Quartz phenocrysts (0,4-0,9mm); Plagioclase; K-feldspar; rare biotite Opaque minerals + rare xenoliths | Muscovite rare micro-patches; Sericite (up to 12%) |
| | Zg-112 | | | |
| | Zg-113 | | | |
| Micro-granite | Zg-02 | fine to medium-grained texture | Amphibole (0,1-0,4mm; rare: less than 3%); Biotite (<0,9mm; 2-4%); Quartz (0,4-3,2mm; less than 40%); K-feldspar (up to 32%); Plagioclase (1-2,9mm; up to 20%); Opaque minerals (<4%); + xenoliths (up to 4,5mm) | Abundant Sericite (up to 10%); chlorite (<3%); rare epidote (<1%) |
| | Zg-03 | | | |
| | Zg-109 | fine grained texture | Amphibole (0,1-0,4mm; rare: less than 3%); Biotite (<0,5mm; 2-4%); Quartz (0,4-0,9mm; less | Abundant Sericite (up to 10%); chlorite (<3%); rare epidote (<1%) |





| | | than 48%); K-feldspar (up to 30%); Plagioclase (0,5-1,4mm; up to 22%); Opaque minerals (<4%) | |

**Table 1: Mineralogical compositions of the studied rocks of the Saghro Group and Ouarzazate Group from the Sirwa massif (Zgounder Mine Region; 33 samples), refer to Fig. 2 for surface sample's locations.**

**Figure 3: Photomicrographs (all in cross-polarized light) of the studied Saghro Group and Ouarzazate Group rocks. (a) Gabbro sample showcasing a medium to coarse- grained locally equigranular texture. (b) An intergranular texture for dolerite. (c) Andesite**
**with a medium grained intergranular texture. (d) Microlitic-porphyritic texture for basaltic-andesite. (e) Sandstone (Zg-132) exhibiting millimetre-scale laminations showcasing darker bands enriched in chlorite, with quartz-muscovite lighter laminations. (f) Microcrystalline texture for rhyolite with embayed quartz phenocrysts. (g) Dolerite exhibiting an intergranular texture . (h) Diabase characterized by an intergranular texture with patches of blue chlorite. (i) Welded ignimbrite with an eutaxitic texture displaying flow textures with feldspars (sanidine) crystals embedded in a glassy fiamme. (j) Quartz diorite with subhedral to**
**anhedral amphiboles and sericitized plagioclase. (k) Medium to locally fine-grained granite, with an anhedral to subhedral equigranular texture.**





## 4.2 Cathodoluminescence (CL) imaging and LA-ICP-MS U-Pb zircon geochronology

### 4.2.1 Tittalt rhyolite (Zg-106)

The rhyolite sample Zg-106 was taken from the Tittalt Formation as the lower unit that represents the Saghro Group in the
Sirwa massif (Thomas et al., 2002, 2004). For context, this rhyolite is represented as a compact and homogenous body of
approximately 46 m length, and 4 to 5 m thick. On the field, it corresponds to a rhyolitic dike crosscutting the mafic to
intermediate units of the Tittalt Formation (Fig. 2). Admittedly, similar subvolcanic activity represented by E-W-trending
rhyolitic dikes and related plugs was previously reported in the Zgounder Mine Region by Pelleter et al., (2016), and attributed
to the Assarrag suite of Thomas et al., (2002). On the outcrop, it has a light pink to brownish taint with visible laminations. U-
Th ratios range between 0.3 and 1. Most of the zircon crystals in this sample are euhedral and moderate in size. CL images
reveal that zircon grains exhibit typical oscillatory zoning with rare and limited dark rim overgrowths (Fig. 4-1a), with no
indication of inherited zircon grains. For filtering the data, only zircons with 95% to 105% concordance were used for age
calculation (Fig. 4-1b). Thus, the weighted mean age for this sample gave 575 ± 3 Ma (MSWD = 1.5; n = 21/40), which is
interpreted as the crystallization age (Fig.4-1c).

### 4.2.2 Granite (Zg-119)

The sample Zg-119 was sampled from the diamond drill hole (DDH) ZG-20-21 (interval in meters: 555,30m –to- 568,50m),
representing the Zgounder Mine exploration program from 2020, and referred to as the 'Deep Intrusion' (Mine terminology).
On hand, the sample corresponds to a fine to medium-grained light grey granite with pink passes. Zircon grains from this
sample are homogenous in size, mostly euhedral to subhedral with some rounded zircon grains. CL images show visible
oscillatory zoning with local moderately fractured zircons showing dark overgrowths (Fig. 4-2a). The discordance filter used
for this sample was the same as for Zg-106; thus, zircons falling outside 95% to 105% concordance were omitted from the age
calculation. A total of high-quality zircon analyses yielded a reliable concordia age of 564 ± 2 Ma (MSWD = 1.6; n = 34/37).
We also calculated a weighted mean age using a broader selection of concordant zircons within error, which gave an equivalent
age of 564 ± 2 Ma (MSWD = 1.1; n = 76/160) (Fig. 4-2b). However, the zircon U-Pb are quite complicated and seem to spread
over ~80 Ma which is likely due to the combined effects of Pb loss and inheritance (Fig. 4. 2-b). Hence both were omitted
from the age calculation. We interpret the youngest tail of ages (~ 520 Ma) which are clearly outside of the main population
reflecting analysis that have been affected by Pb loss (Sharman and Malkowski, 2024). Whilst oldest zircons are probably
inherited (~ 620 Ma), and referring to a detrital signature. However, the analyzed detrital sample (Zg-132), that represents the
sedimentary succession in which this granite is intruded has no zircons of this age (see supplementary table 5 for Zg-132).
Moreover, the inheritance signature cannot be supported as no xenocrystic zircons were observed in the CL images. Overall,
as both ages are statistically identical, we interpret the 564 ± 2 Ma as the robust crystallization age for the granite (Fig. 4-2c).





### 4.2.3 Sandstone (Zg-132)

This sample was collected from the Ag-Hg hosting sedimentary units from the Imghi Formation of Saghro Group near the Zgounder Mine entrance (Fig. 2). It is a fine to medium-grained greenish sandstone. On the outcrop, it follows the E-W-trending direction of the Imghi Formation, dipping at 70° to the south. For Thomas et al., (2002), the Imghi Formation represents the lower thick beds of greywacke/turbiditic section of Saghro Group, and seems to represent a typical flysch succession, which is interpreted as the primary sedimentary fill of a back-arc basin associated with the Khzama ophiolite complex. Most zircons from this sample are uniformly rounded and fragmented. They are distinguishable by their low CL signal with some being preferentially zoned (Fig. 4-3a). The age calculation utilized a filtered dataset, specifically incorporating only zircons that fell within the 95% to 105% concordance range. A total of 139 analyses were obtained on zircons from this sample, of which 91 are concordant (Fig. 4-3b). The ages distribution is dominated by one single population. For the detrital zircons, one uniform peak at around 2100 Ma, with one zircon recording an age at 3700 Ma (Fig. 4-3c).





**Figure 4: Cathodoluminescence (CL) images for selected zircons, U-Pb concordia and Pb$^{206}$ / U$^{238}$ weighted mean diagrams for samples (1) Zg-106 (rhyolite), and (2) Zg-119 (granite); along with U-Pb concordia diagram and KDE for (3) Zg-132 (sandstone). Error ellipses are plotted at 2σ. Ages were calculated using Isoplot R 1.0 (Vermeesch, 2018).**

### 4.3 Whole-rock geochemistry

Due to samples distribution and geological diversity, the selected samples are organized in respect to their lithostratigraphic position, and hence treated accordingly. The whole-rock data for 32 samples are listed in supplementary table 4. For a start, we assessed the behavior of mobile elements and how it relates with the formed alteration minerals, using the diagram of Large




et al., (2001); and samples are plotted in (Fig. 5). The rock material loss of ignition (LOI) at 1100 °C is low for most samples (less than 2 wt. %), except for the mafic and intermediate terms of Saghro Group where it is close to 4 wt. % (e.g. basalt, gabbro and andesite samples). On plotting our data in this diagram, a homogenous hydrothermal trend is clearly defined for all samples except for sample (Zg-106). Its trend is more directed towards sericite alteration, typical of a distal ore environment

(Large et al., 2001). Overall, the plot confirms that most of our samples plot in the domain of '''Least altered rocks'''.

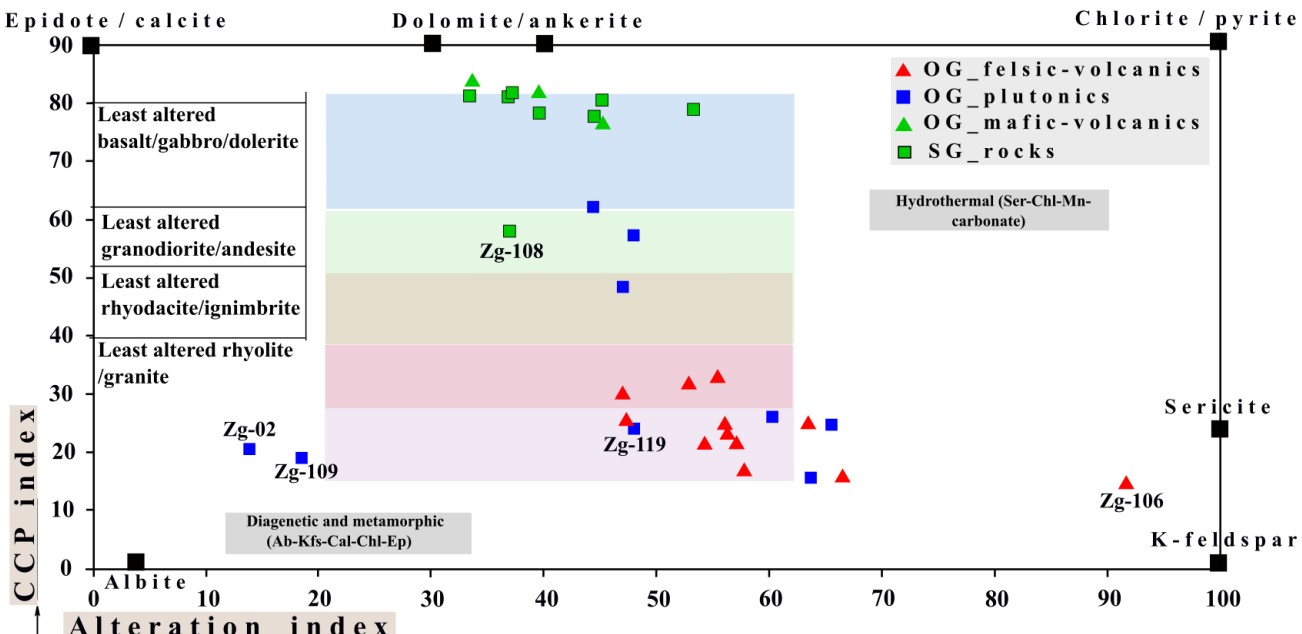

**Figure 5:** Plots of the studied rocks on the alteration box diagram (Large et al., 2001). Alteration Index (AI) = 100 x $(K_2O + MgO)$ / $(K_2O + MgO + Na_2O + CaO)$, Chlorite-carbonate-pyrite index (CCPI) = 100 x $(MgO + FeO)$ / $(MgO + FeO + Na_2O + K_2O)$.

**4.3.1 The Saghro Group rocks**

The analyzed Saghro Group rocks belong to the Tittalt Formation of the Saghro Group in the study area (Fig. 1C; Fig. 2; Thomas et al., 2002, 2004). Geochemically, they define a narrow range of compositions, with $SiO_2$ contents from 48.85 wt.% to 52.87 wt.%, except for sample (Zg-108) with $SiO_2$ of 61.15 wt.%. They contain 1.08 – 7.45 wt.% MgO; 13.44 – 15.65 wt.% $Al_2O_3$; 2.82 – 3.95 wt.%, $Na_2O$ and 0.47 – 3.07 wt.% $K_2O$. Based on the Nb/Y vs $Zr/TiO_2$ diagram (Fig. 6a), except for one sample (Zg-108) with andesitic composition, all samples plot within a sub-alkaline basalt; similar to contemporaneous mafic-

intermediate rocks from the Saghro (Errami et al., 2009) and Sirwa inliers (Thomas et al., 2002). Most rocks are metaluminous except for one sample (Zg-006) which is peraluminous (Fig. 6b). In addition, the Saghro Group rocks display a calc-alkaline signature (Fig. 6c).

Using rare earth elements (REE) (Fig. 6d), the Saghro Group samples show a total REE varying between 60.94 to 231.77 ppm with coherent patterns characterized by a slight LREEs enrichments and HREE depletion with $(La/Yb)_N$ ratios ranging from

1.83 to 6.42. Almost all samples have slight/no important negative Eu anomalies (δEu = 0.88–0.98) to less positive anomaly





for one sample of gabbro (Zg-105; Eu/Eu*)N = 1.28; Fig. 6d). For the multi-elements diagram (Fig. 6e), the Saghro Group rocks exhibit an enrichment of large ion lithophile elements (LILE, like Rb, Ba, K) over high field strength elements (HFSE) with negative anomalies in Nb, Ta, and a prominent positive Pb anomaly. They do compare in the most part with basaltic rocks from the Saghro Group of Kelâat M'gouna inlier (Benziane et al., 2008; Fekkak et al., 2001), and the early gabbro sample

from Sirwa (Touil et al., 2008; Fig. 6d).

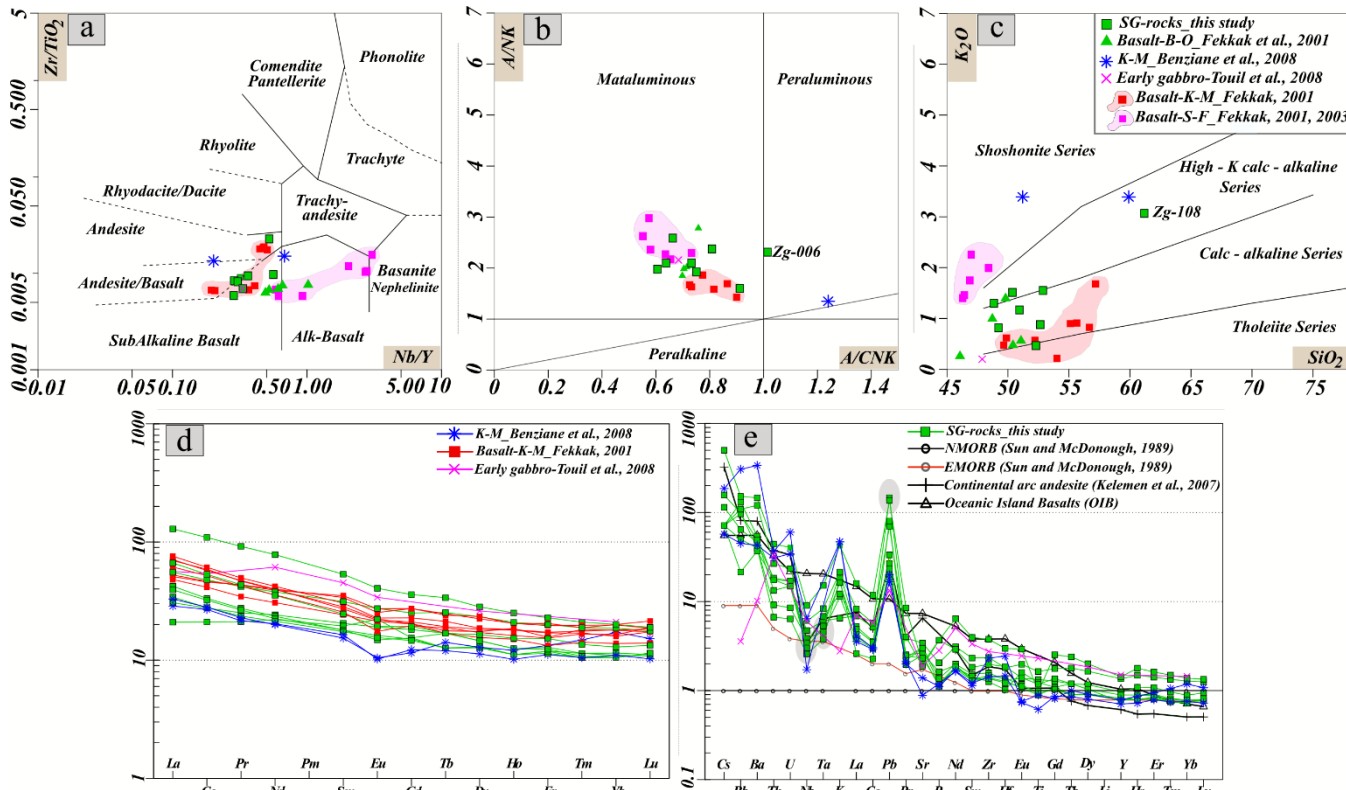

**Figure 6: Saghro Group:(a) Nb/Y versus Zr/TiO₂ (Winchester and Floyd, 1977). (b) Plot of A/NK vs. A/CNK [A/CNK = molar ratio Al₂O₃ / (CaO + Na₂O + K₂O) and A/NK = molar ratio Al2O3 / (Na₂O + K₂O)](Shand, 1943). (c) K₂O versus SiO₂. (d) Chondrite-normalized REE diagram of Boynton, (1984). (e) NMORB normalized diagram; normalization values are sourced from Sun and**

**McDonough, (1989). SG samples are plotted against reference rocks from the literature. Abbreviations: B-O : Boumalne; K-M: Kelâat M'gouna; S-F: Sidi Flah.**

### 4.3.2 The Ouarzazate Group rocks

The studied samples representing the Ouarzazate Group are composed of both volcanic and plutonic rocks. The felsic volcanic and plutonic samples have high contents in SiO₂ (79.89 – 61.16 wt.%), Al₂O₃ (15.18 – 10.83 wt.%), and CaO (4.42 – 0.1 wt.%)

and low MgO (2.83 – 0.06 wt.%), Ni (60 - 20 ppm) and Co content (19 - 1 ppm), except for one granite (Zg-119) that have relatively high Co content of 109 ppm. Moreover, they have a Na₂O + K₂O = 8.97 to 6.31 with a K₂O/Na₂O ratio of 20.75 to 0.14. Rock types for felsic volcanics are as follows: ignimbrites are represented as rhyodacite and dacite, while rhyolites and rhyolitic-ignimbrites plot inside the rhyolite field (Fig. 7a). Furthermore, the plutonic rocks occupy the granite field, with three




distinctive granodiorites (not shown). The whole set consist of high-K calc-alkaline to shoshonitic rocks (Fig. 7b), and are
peraluminous, except for the granodiorite samples that exhibit a metaluminous character (Fig. 7c). Overall, all samples plot
within the calc-alkaline and alkali-calcic slightly migrating towards alkaline field (Fig.7d).

Mafic volcanics (e.g. dolerite) have moderate $SiO_2$ contents of (51.56 – 50.58 wt.%), $Al_2O_3$ (14.69 – 13.9 wt.%), moderate
CaO (7.67 – 5.61 wt.%) and moderate MgO (5.3 – 4.56 wt.%). $K_2O/Na_2O$ ratio values are 0.73 to 0.26, along with $Na_2O+K_2O$
ratio = 5.64 to 3.58 (supplementary table 4). They are sub-alkaline basalts (Fig. 7a), with a calc-alkaline signature (Fig. 7b),
and a metaluminous character (Fig. 7c).

In the REE diagram (Fig. 7e), all felsic volcanic and plutonic samples show homogeneous patterns, with enriched LREEs over
HREEs. $(La/Yb)_N$ ratio varies between 17.93 and 2.1, except for one rhyolitic sample Zg-106 with $(La/Yb)_N$ of 0.89. All
samples have high Eu negative anomaly (Eu/Eu* = 0.61 – 0.34, except for the same Zg-106 sample, that exhibits a prominent
Eu anomaly with Eu/Eu* = 0.04, controlled by advanced plagioclase crystallization. In the multi-element spider diagram, all
samples display enrichments in LILEs (Cs, Rb, Ba, Th, and U) and depletion in HFSEs (Zr, Nb, Hf and Ta), relative to Primitive
Mantle (Fig. 7f). Plus, samples show a prominent negative anomaly in Nb, Ta, P and Ti along a positive anomaly in Pb, Ba
and Th, except for sample Zg-106 that lacks negative Nb anomaly.

For the mafic volcanics, the REE normalized to chondrite (Fig. 7e) show moderate LREE enrichments over HREE with less
fractional patterns $((La/Yb)_N$ = 6.22 to 5.94) and very low/absent Eu anomaly (Eu/Eu* = 0.98 to 0.88). However, our dolerites
are distinguished by their homogenous pattern with a prominent Pb enrichment, with less significant Sr anomaly (Fig. 7f). Our
dolerites resemble those of the Tifnout Valley (TV), Group A and B, and Zaghar mafic dikes from the Sirwa massif by
Belkacim et al., (2017); Touil et al., (1999); Toummite et al., (2013), respectively. Even so, minor differences related to
prominent negative Nb and Ti anomaly are observed for reference samples compared to our dolerites' positive Ti anomaly
(Fig. 7f). However, they differ from the Bas Drâa inlier mafic-intermediate bodies, which lack Pb enrichment and Sr depletion
Karaoui et al., (2014).





**Figure 7: Ouarzazate Group: (a) Nb/Y versus Zr/TiO₂ (Winchester and Floyd, 1977). (b) Th vs Co plot of Hastie et al., (2007). (c) Plot of A/NK vs. A/CNK of Shand, (1943). (d) MALI diagram of Frost et al., (2001). (e) Chondrite-normalized REE diagram of Boynton, (1984). (f) Primitive mantle normalized diagram of Sun and McDonough, (1989). OG samples are plotted against reference rocks from the literature. Abbreviations: TV: Tifnoute Valley; BD: Bas Draâ.**





## 4.4 Significance of Sm-Nd isotopic data

Sm-Nd isotopic studies of seven representative samples covering the Ediacaran record of the Sirwa massif reveal petrogenesis processes and orogenic evolution between the 630 to 538 Ma evolution of the Anti-Atlas belt post-Bou Azzer-Sirwa ophiolite accretion. Results are listed in table 2.

| Lithology | Sample code | Sm (ppm) | Nd (ppm) | 147Sm/144Nd | 143Nd/144Nd | ± 2 SE | ɛNd(0) | ɛNd(570) | TDM (Goldstein) Ma | TDM Ga |
|---|---|---|---|---|---|---|---|---|---|---|
| **Ouarzazate Group rocks** | | | | | | | | | | |
| Granodiorite | Zg-09 | 10.23 | 50.70 | 0.1220 | 0.512414 | 0.000005 | - 4,4 | + 1,1 | 1252,8 | 1,24 |
| Dolerite | Zg-14 | 6.518 | 28.99 | 0.1359 | 0.512390 | 0.000010 | - 4,8 | - 0,4 | 1526,7 | 1,51 |
| Rhyolite | Zg-106 | 11.50 | 32.75 | 0.2124 | 0.512693 | 0.000007 | + 1,1 | - 0,1 | 0 | N/A |
| Rhyolite | Zg-117 | 10.69 | 50.59 | 0.1278 | 0.512332 | 0.000011 | - 6,0 | - 0,9 | 1483,7 | 1,47 |
| Granite | Zg-119 | 9.106 | 42.92 | 0.1283 | 0.512381 | 0.000009 | - 5,0 | 0,0 | 1404,8 | 1,39 |
| **Saghro Group rocks** | | | | | | | | | | |
| Gabbro | Zg-105 | 3.739 | 13.27 | 0.1704 | 0.512762 | 0.000011 | + 2,4 | + 4,5 | 1431,6 | N/A |
| Andesite | Zg-108 | 8.943 | 40.03 | 0.1351 | 0.512550 | 0.000007 | - 1,7 | + 3,2 | 1197,3 | 1,19 |


**Table 2: Sm-Nd isotopic data for the analyzed Saghro Group and Ouarzazate Group rocks. TDM is the Depleted Mantle Age in Ga calculated using the linear model of Goldstein et al., (1984); and is not calculated for samples with 147Sm/144Nd ratios greater than 0.145 (e.g. Zg-106, Zg-105). Abbreviations: ± 2 SE: Standard Errors (Uncertainty in Nd isotopic composition); N/A : Not Available.**

Based on field relationships, An angular unconformity (Errami et al., 2009; Thomas et al., 2002), and a significant
sedimentary/magmatic gap exist between the 630 - 600 Ma Saghro Group and the overlaying 570 - 538 Ma Ouarzazate Group (Errami et al., 2021a). We have used the 620 Ma as a reference age for ɛNd analyses of Saghro Group samples (Zg-105, and Zg-108). They show narrow ($^{143}$Nd /$^{144}$Nd) 620 Ma ratios, from 0.512001 to 0.512070, and positive ɛNd (at 620 Ma) values between + 3.2 to + 4.5. Additionally, the TDM model ages (Fig. 8a), ranging from 1431 – 1197 Ma, exceed the individual maximum depositional ages of 630 to 600 Ma, and the crystallization ages of pre to contemporaneous igneous intrusions found
elsewhere in the Anti-Atlas belt that fall within the bracket of 640 Ma to 580 Ma, respectively (Errami et al., 2021a; O'Connor et al., 2010; Liégeois et al., 2006; Gasquet et al., 2005; Mrini, 1993). Overall, the Saghro Group samples show a mixed origin with ɛNd (620 Ma) ranging from + 3.2 to + 4.5, indicating a blend of mantle-derived magma and moderate contribution of an old crust (Paleoproterozoic ?). This contrasts the basaltic rocks studied in the Saghro region by Errami et al., (2009), for which ɛNd (at 640 Ma) vary between + 7.63 to + 8.08, suggesting a juvenile source, with no Paleoproterozoic influence.
Consequently, authors argued that the Saghro Group's sedimentary and volcanic deposits attest to an active back-arc basin, with the arc itself located north of the Saghro mountains in the Saghro inlier (Errami et al., 2009).

Ouarzazate Group rocks exhibit a mostly uniform distribution with ($^{143}$Nd/$^{144}$Nd) 570 Ma values from 0.511855 to 0.511958. These ratios are close to or slightly lower than the Chondritic Uniform Reservoir (CHUR) values. Moreover, for all samples, the ɛNd570 values range from - 0.9 to + 1.1. Volcanic rocks show negative ɛNd570 values: -0.9 (Tadmant rhyolite), - 0.1
(Ouarzazate Group rhyolite), and - 0.4 (dolerite). Granitoids have ɛNd570 values of + 0.0 to + 1.1 for Zg-119 and Zg-09, respectively. This range in values suggests mixing or variable degrees of interaction between a mantle source and an older




crustal component. In addition, the TDM model ages for all of these samples range from 1526 to 1252 Ma (Fig. 8c), indicating a Mesoproterozoic affinity, even though the effect is limited. Admitting that Mesoproterozoic rocks are scarce to absent in the Anti-Atlas, these Proterozoic TDM ages, in conjunction with the mixed (near zero) εNd (570 Ma) values for both volcanic and

plutonic rocks, strongly suggest that the magma for the Ouarzazate Group are likely to represent mixed values involving a Pan-African juvenile mantle together with moderate but discernible contribution from an older crustal material of Mesoproterozoic in age, which itself incorporated some even older (Paleoproterozoic ?) material (Baidada et al., 2017; El Bahat et al., 2017; Blein et al., 2014b; Gasquet et al., 2005; Thomas et al., 2002) (Fig. 8d). All in all, these results do confirm the presence of an old cratonic basement beneath the Central Anti-Atlas.

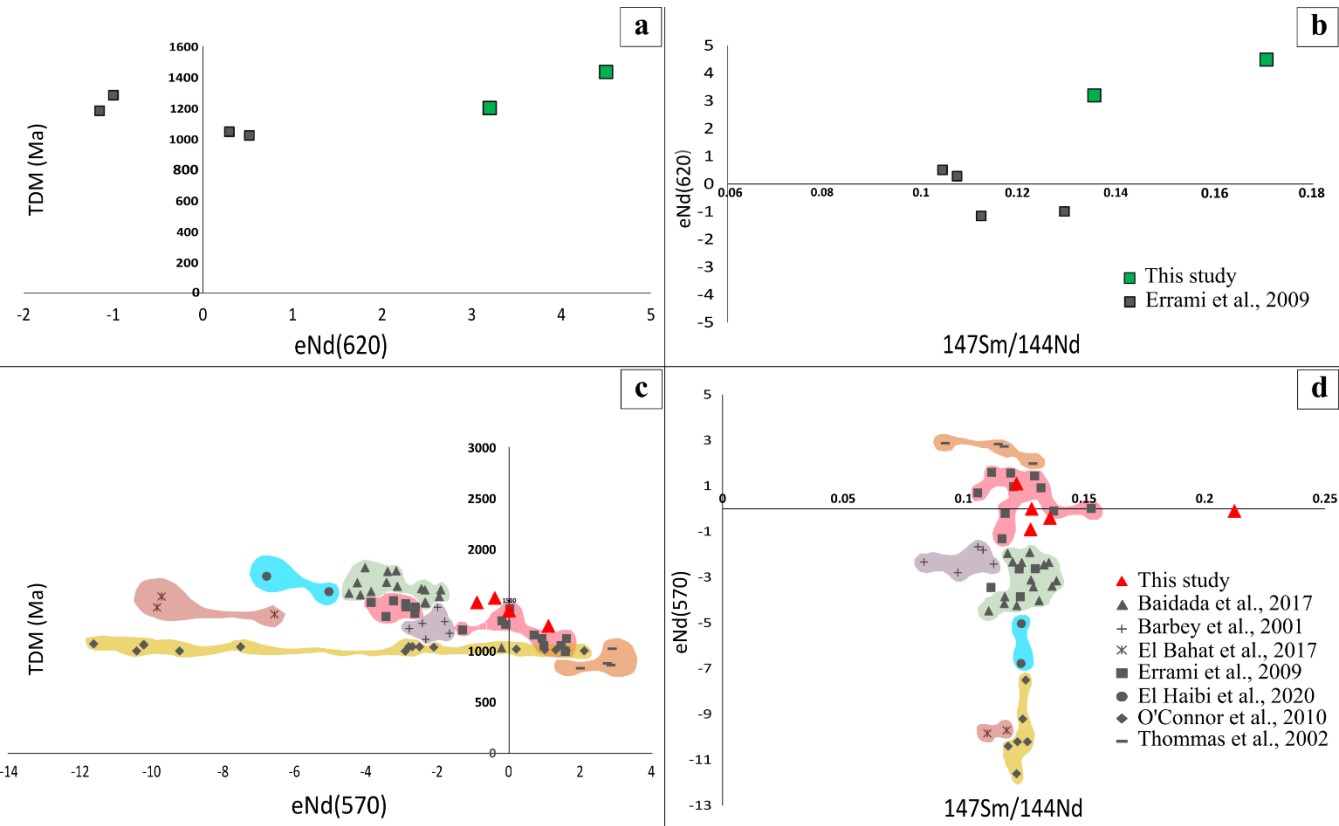


**Figure 8: (a) εNd$_{620}$ vs TDM model ages for SG. (b) εNd$_{620}$ vs $^{147}$Sm/$^{144}$Nd diagram for SG. (c) εNd$_{570}$ vs TDM model ages for OG. (d) εNd$_{570}$ vs $^{147}$Sm/$^{144}$Nd diagram for OG rocks. All samples are compared with selected and available reference rocks from the literature.**





## 5 Discussions

### 5.1 Petrogenesis

#### 5.1.1 The Saghro Group rocks

The Saghro Group rocks consist of basaltic lava flows interbedded with marine, low-grade metasediments. The geochemical fingerprint of the mafic rocks is well-suited for investigating the orogenic processes involved in magma genesis (Pearce, 1983; Wood, 1980). However, it is crucial to evaluate the effects of alteration, metamorphism, and crustal contamination before exploring the magmatic evolution.

- **Alteration effect**

The analyzed samples exhibit loss on ignition (LOI) values of 2 to 4 wt.%, implying major elements and LILE mobility during alteration process (supplementary table 4). Saghro Group basalt samples display scattered large-ion lithophile elements (LILEs; e.g., K, Rb, Na, Sr, Pb) vs. Zr, suggesting alteration variances. In contrast, binary plots of high-field-strength elements (HFSEs; e.g., Nb, Ta, Ti, Hf) and rare earth elements (REEs) against Zr exhibit consistent linear trends, reflecting their immobile nature (Pearce, 1996; supplementary table 4). All samples from this Group are classified as minimally altered basalts (Large et al., 2001) (Fig. 5).

- **Fractional crystallization and crustal contamination**

The analyzed samples predominantly consist of basalts displaying negative correlations for $Al_2O_3$, CaO, MgO, $Fe_2O_3$, and $TiO_2$ vs. Zr, and a positive correlation for $P_2O_5$, indicating fractional crystallization during magma evolution. The Mg# values (12.13 to 43.26), reflect early fractionation of ferromagnesian minerals, and suggest that the Saghro Group samples are not derived from a primitive melt. Furthermore, the absence of a significant Eu anomaly (δEu = 0.88–1.28) indicates limited plagioclase fractionation (Fig. 6d). The lack of a negative Eu anomaly, despite the evolved nature of the magmas (low Mg#), is likely due to the mixing of evolved basaltic magmas (low Mg#), with more primitive magmas that lacked plagioclase fractionation.

In a contamination-sensitive trace element diagram (Th/Ce and Th/La) (Fig. 9a), our samples show ratios of 0.047 to 0.080 and 0.115 to 0.172, respectively. The trend suggest that crustal contamination played a significant role in magma genesis (Taylor and McLennan, 1995). Additionally, the low Ce/Pb ratio (~ 4.3) further highlights continental crustal influence (Hofmann et al., 1986; supplementary table 4 for calculation). Further, a binary mixing model with Nd isotopic data (supplementary table 4 for calculation; De Paolo, 1981), interprets the Saghro Group basalts as formed from the mixing of a magmatic melt contaminated by 17 – 18% Paleoproterozoic continental crust (εNd620 = - 16; Ennih and Liégeois, 2008) and 82 – 83% primitive mantle (εNd620 = + 8; Errami et al., 2009). This is further supported by the negative correlation of εNd values with $SiO_2$, and relatively low εNd values (+ 3.2 to + 4.5), compared to contemporary mantle-derived Saghro inlier basalts (εNd = + 8; Errami et al., 2009).

However, the observed enrichment may stem from either continental crust contamination or mantle source enrichment. The Th/Ce and Th/La ratios for our samples plot linearly near the MORB reservoir, far from the upper continental crust (UCC)



field, suggesting limited direct crustal input (Fig. 9a). Moreover, ratios such as (Sm/Yb)N versus (Nb/La)N (Safonova et al., 2016), sensitive to the nature of the mantle source, range from 1.64 to 2.74 and 0.50 to 0.91, respectively. These values place the Saghro Group samples in a transitional domain between MORB and IAB, closer to the field of back-arc basalts (BAB)

(Fig. 9b). Similarly, Th/Nb and Ta/Nd ratios (0.12 – 0.30 and 0.03 – 0.06, respectively) indicate a transitional composition between MORB and the mafic lower continental crust (MLCC) (Aldanmaz et al., 2008; Fig. 9c), in alignment with the source diagram (Laurent et al., 2014; Fig. 9d), where all the Saghro Group rocks derived from fractional crystallization and contamination/assimilation of primary mafic lithologies straddling the boundary between high-K to low-K mafic rocks. Overall, the Saghro Group samples originate from the melting of an enriched source transitional between MORB and IAB,

followed by fractional crystallization involving ferromagnesian minerals and assimilation of lower Paleoproterozoic continental crust.

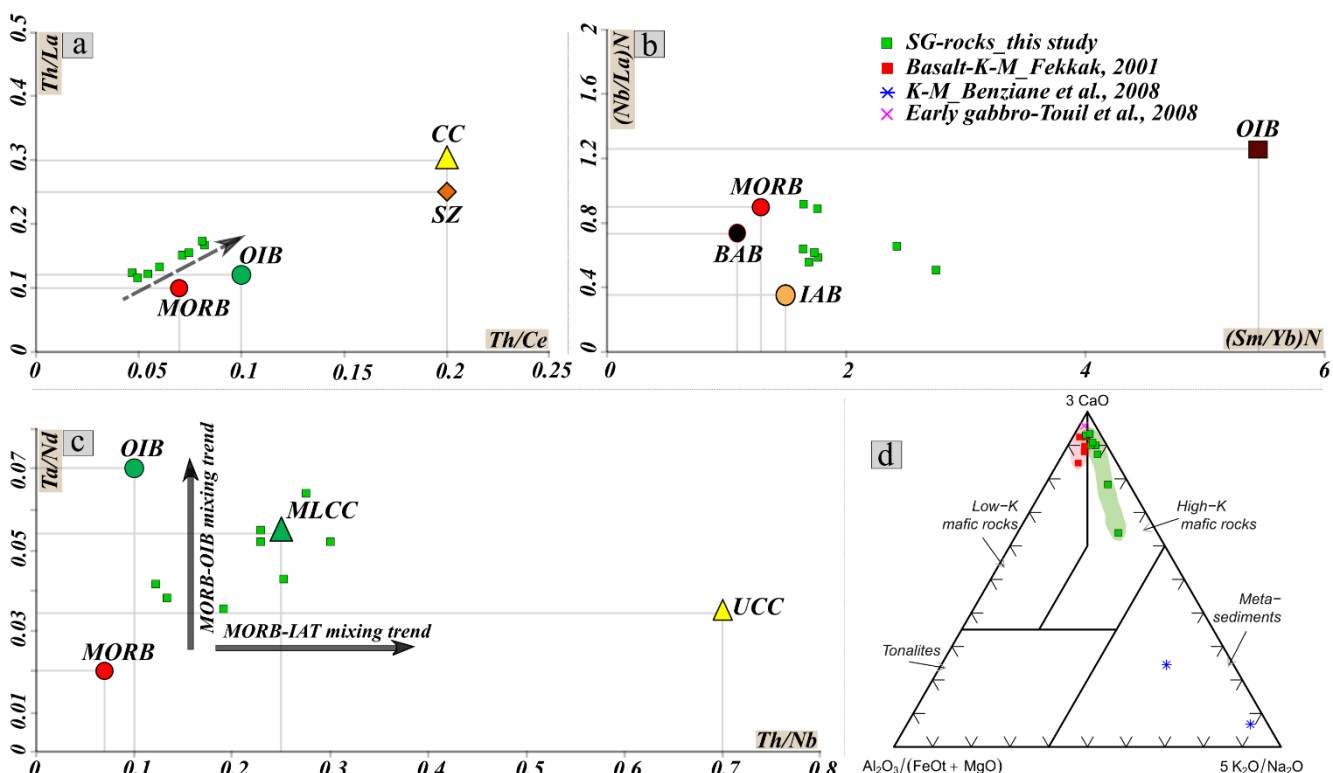

**Figure 9: Saghro Group: (a) Th/Ce vs Th/La ratios for SG (Taylor and Mc Lennan, 1995). (b) plot of (Sm/Yb)N vs (Nb/La)N (Safonova et al., 2016). (c) Th/Nb vs Ta/Nd plot (Aldanmaz et al., 2008). (d) The source diagram of Laurent et al., (2014).**
**Abbreviations: K-M: Kelâat M'gouna.**

**5.1.2 The Ouarzazate Group rocks**

▪ **Fractional crystallization and crustal contamination**



The Ouarzazate Group's volcanic and plutonic rocks exhibit chemical evolution that can be explained by fractional crystallization, with decreasing Al2O3, Fe2O3, MgO, CaO, TiO2, and P2O5 vs. rising SiO2. Major elements vs. Zr show no

significant correlations (not shown: see supplementary table 4). Selected trace elements, such as Sr, decrease with rising SiO2, typical of plagioclase crystallization. Interestingly, Some trace elements (e.g. Cr, Ni, Ba, Rb, Zr) show disruptions in their patterns with increasing SiO2. Ni uniformly declines then sharply increases, Cr rises steadily then increases, while Ba, Zr, and Rb rise then decline, particularly at 70 wt.% SiO2 (Fig. 10a). These patterns are likely due to a magma recharge event affecting pre-magma's crystallization (Fig. 10a).

Felsic volcanics and plutonics compensate for Sr and P depletion due to plagioclase and apatite fractionation (Fig. 7f). A prominent negative Eu anomaly in rhyolite Zg-106 indicates early calcic phase crystallization, while the absence of an Eu anomaly in dolerites suggests limited plagioclase involvement (Fig. 7e). Additionally, their Pb enrichment indicates crust assimilation (Fig. 7f). The high La (16.4 to 133 ppm) and Ce (33.8 to 284 ppm) values in the Ouarzazate Group samples suggest interaction between parental magma (s) and crust material, indicating enrichment due to either assimilation of

continental crust or a combination of assimilation and fractional crystallization during magma ascent (De Paolo, 1981). Further, fractional crystallization relates to correlations between Zr and Th/Nb ratios (Fig. 10b), as most of the Ouarzazate Group felsic volcanics and plutonics show similar Th/Nb and Zr trends, indicating significant assimilation and fractional crystallization in magma evolution. Some plutonics, however, show a limited Th/Nb range, suggesting bulk assimilation control (Fig. 10b). This is supported by the La/Sm versus La diagram (Fig. 10c) indicating source heterogeneity, despite fractional crystallization's

dominance. The Y/Nb versus SiO2 wt.% (Fig. 10d) confirms a crustal source with varying differentiation. Depletions in P, Nb, and Ti in Ouarzazate Group rocks (Fig. 7f) suggest crustal contamination and hydrous metasomatism, with P anomalies linked to Nb and Ti depletions (Campbell et al., 1994). In addition, the depletion of HFSEs like Nb and Ti might also indicate magmatic arc signatures and possible crustal contamination during magma processes (Wilson, 1989). All samples show slight K enrichments (Fig. 7f), likely related to fluids from subducted sediments (Beraaouz et al., 2004).








**Figure 10: Ouarzazate Group: (a) Harker variation diagrams for selected trace elements (e.g. Sr, Cr, Ni, Ba, Rb, Zr) for the OG rocks. (b) plot of Th/Nb vs Zr. (c) La/Sm vs La plot. (d) Y/Nb vs increasing SiO$_2$, (Eby, 1990) for the Y/Nb =1.2 discriminating value. Abbreviations: BA: Bulk Assimilation; AFC: Assimilation and Fractional Crystallization; FC: Fractional Crystallization.**

- ▪ **Magma types and source characteristics**

The coherent negative anomalies in Nb, P, and Ti in the Ouarzazate Group samples might suggest an exclusive crustal origin,

with minor discrepancies due to advanced degrees of differentiation. These features may also arise from low partial melting



of metasomatized mantle contaminated by continental material (O'Reilly and Griffin, 2013), as supported by εNd values (- 0.9 to + 1.1) indicating juvenile magma contributions. Indeed, trace elements and isotopic compositions can clarify subduction input versus crustal contamination. The Ouarzazate Group rocks show significant LILE and LREE enrichment with Nb-Ta

depletion, indicating a metasomatized lithospheric mantle from subduction or assimilation of enriched continental crust (Fig. 7-f). Furthermore, the Th/Yb versus Ta/Yb diagram suggests that the magma source originated from an E-MORB-type mantle, evolving through contamination and Assimilation-Fractional Crystallization, as evidenced by the trend of the samples toward the average upper continental crust (Fig. 12a). This interpretation is consistent with the moderate εNd values and TDM ages that reflect contributions from both mantle and continental crustal components, highlighting the importance of crustal

contribution over sediment zone enrichment (Fig. 12b). All in all, the primary magma responsible for the genesis of the Ouarzazate Group rocks sourced from an enriched continental lithospheric mantle, previously underwent metasomatism by fluids from former Neoproterozoic subduction.

## 5.2 Geodynamic implications and regional correlations

### 5.2.1 The Saghro Group

Previously, the Saghro Group basin was interpreted as consisting of distal deep marine sediments equivalent to the platform series of Tachdamt Group, formed during the initial rifting associated with the Rodinia breakup (Thomas et al., 2002). However, recent zircon data from the Saghro Group sediments in the Sirwa and Saghro inliers suggest its deposition, post the Bou Azzer–Sirwa ophiolite accretion, during the main Pan-African orogeny (Abati et al., 2010; Errami et al., 2009; Liégeois et al., 2006).

Basaltic samples from the Sirwa inlier (this study) show enrichment in large ion lithophile elements (LILE) and light rare earth elements (LREE) relative to high field strength elements (HFSE) and heavy rare earth elements (HREE). They also exhibit pronounced negative anomalies in Nb, Ta, and Ti, suggesting a subduction zone fluids influence (Wilson, 1989). Despite this, the basalts feature high Zr and relatively low Y concentrations, classifying them as within-plate basalts (WPB) (Fig. 11a). This dual geochemical signature, blending within-plate and active margin characteristics, is further evidenced by their Ti/Y and

Zr/Y ratios which plot in transitional field in the geotectonic discrimination diagram (Ti/Y versus Zr/Y; Pearce and Gale, (1977), not shown). Indeed, the coexistence of geotectonic signatures from both active continental margins and within-plate settings is characteristic of magmas generated during back-arc extension (Shinjo and Kato, 2000).

In this context, the subduction-related signature is acquired through mantle metasomatism by sediment/fluid-derived components during arc activity, while the within-plate signature emerges later during back-arc extension processes. This

interpretation aligns with the geochemical characteristics of the Saghro Group basalts, which plot within the fields of mid-ocean ridge basalt (MORB) and back-arc basin basalt (BABB) (Fig. 11b). Moreover, the active margin signature is supported by high Th/Nb and low U/Th ratios in the Saghro Group basalts (refer to supplementary table 4). Further, the influence of





subduction-related components is further highlighted by the ThN vs NbN ratios, with the basalts plotting above the MORB-OIB array (Fig. 11c). This pattern indicates a mantle source enriched during continental arc processes (Pearce, 1983).

At a regional scale, basaltic flows have been described as interbedded within the sediments of the Saghro Group in the Saghro massif (Errami et al., 2009; Fekkak et al., 2003, 2001). They exhibit diverse geochemical signatures. For a start, the basalts in Sidi Flah (SF) are of Initial Rift Tholeiites (IRT) and Ocean Island Basalts (OIB) character, while those from the Kelâat M'gouna (KM) inlier are Nb-depleted. In contrast, the Anou N'Izem basalts in the Boumalne (BO) inlier possess typical Mid-Ocean Ridge Basalt (MORB) signatures. U-Pb dating of detrital zircons from the Saghro Group in various inliers suggests that

this unit was deposited between 640 - 600 Ma across the entire Central-Eastern Anti-Atlas region (Errami et al., 2021a; Abati et al., 2010) (Fig. 13A). This geochemical variation may reflect a single geodynamic event. Specifically, during the evolution of a back-arc basin, the earliest magmas generated often exhibit magmatic arc characteristics due to the proximity of the back-arc to the arc itself. As the back-arc basin evolves and the magma source becomes further removed from the metasomatised mantle, the magmas acquire within-plate geochemical signatures (Vasey et al., 2021; Saunders and Tarney, 1984). Overall,

basaltic magmatism in the Saghro Group fits this evolutionary framework. For instance, calc-alkaline basalts from the Sirwa inlier (Thomas et al., 2002; This study), reflect initial stages in back-arc formation. Conversely, and higher in the stratigraphy; IRT, OIB, and MORB-type basalts indicate later stages in back-arc evolution. This interpretation is consistent with sediment geochemistry suggesting back-arc basin deposition (Ouguir et al., 1996). Furthermore, this scenario aligns with the paleogeographic position of the Anti-Atlas during the Ediacaran. During that time, the Anti-Atlas region was dissected in a

northward direction (present coordinates) due to south-dipping Cadomian subduction (Fig. 13) (Rojo-Pérez et al., 2024; Stern, 2024; Errami et al., 2021a; Linneman et al., 2014).



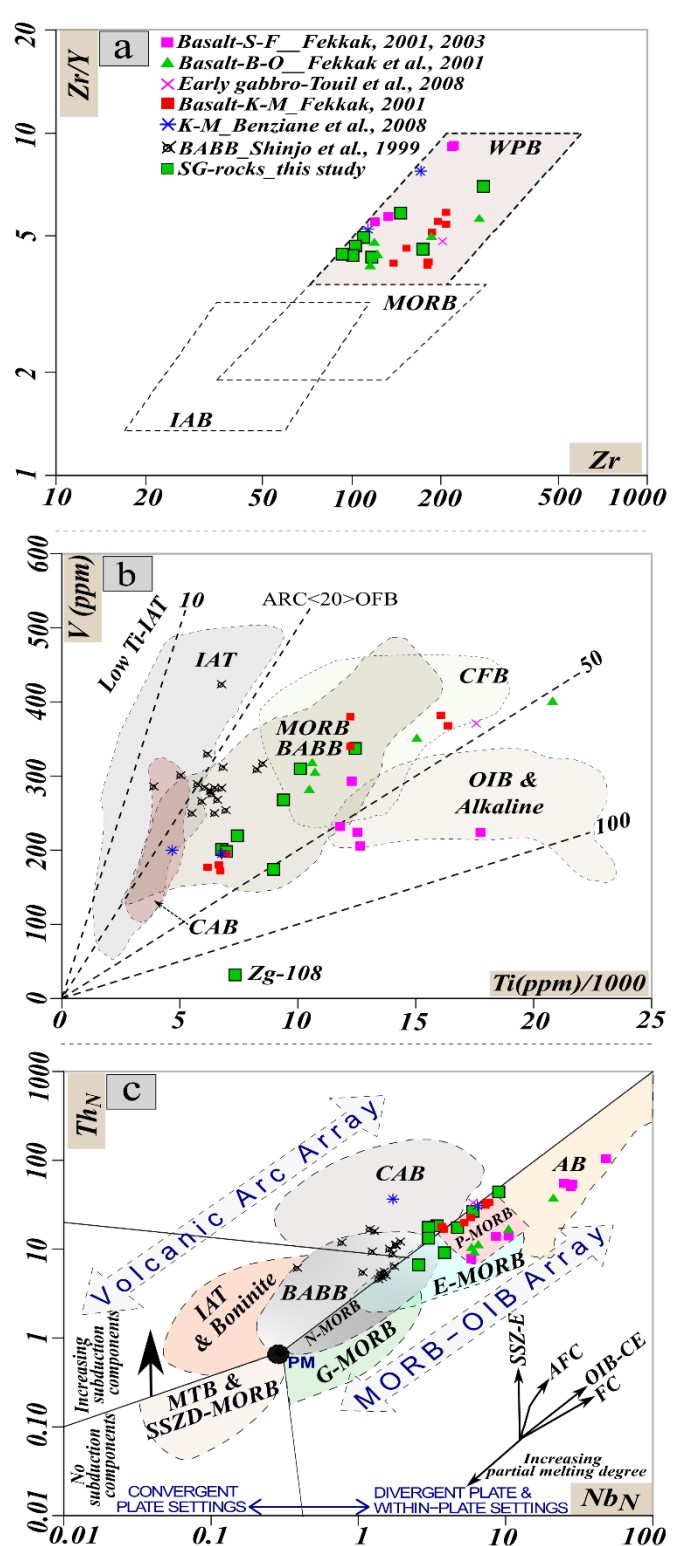





**Figure 11: Saghro Group: (a) Zr/Y vs Zr plot for basalts of SG (Pearce and Norry, 1979). (b) V vs Ti/1000 plot (Shervais, 1982). (c) plot of $Th_N$ vs $Nb_N$ ratios (Saccani, 2015). Abbreviations: B-O : Boumalne; K-M: Kelâat M'gouna; S-F: Sidi Flah; BABB: Back-arc** 495 **basin basalts.**

### 5.2.2 The Ouarzazate Group

On the tectonic discrimination diagrams (Fig. 12b-c), the Ouarzazate Group rocks display geochemical characteristics ranging from dominating within-plate to minor active continental margins (oceanic island arcs). Plus, the high Th/Yb ratios > 1 for all rocks refer to a metasomatised mantle source in subduction zone and/or crustal contamination. All in all, the whole accounts

for a post-collisional setting (Fig. 12c), involving partial melting of a pre-existing lithospheric source, either a mantle or crust. Hence, controlling the recharge in $K_2O$ and LILE (e.g. Rb, Ba) contents for the majority of our samples (Fig. 11a). Additionally, the alkali-calcic to high-K calc-alkaline signature of our samples is in fact typical to post-collisional events (Liégeois et al., 1998, and reference therein).

The Ediacaran magmatism of the Ouarzazate Group is still debated as being fully post-collisional and linked to asthenospheric

rise (upwelling) beneath the WAC during its metacratonic evolution (Belkacim et al., 2017; Gasquet et al., 2008, 2005; Liégeois et al., 2006; Thomas et al., 2002), or related to subduction (Walsh et al., 2012; Benziane, 2007; El Baghdadi et al., 2003), or even representing the Iapetus Ocean opening with ties to the Ediacaran Central Iapetus Magmatic Province (CIMP; Youbi et al., 2020). Nonetheless, subduction-related features can arise in a non-subduction settings without coeval subduction (Morris et al., 2000; M'arquez et al., 1999; Cousens, 1996; Hooper et al., 1995).

Regional correlations with similar magmatism lead us to attribute the Ouarzazate Group to a post-collisional event also exemplified in numerous inliers of the Anti-Atlas (Fig. 13C-D) (Yajioui et al., 2020; Karaoui et al., 2015; Linneman et al., 2014; Toummite et al., 2013; Walsh et al., 2012; Gasquet et al., 2005, 2004; Thomas et al., 2002).



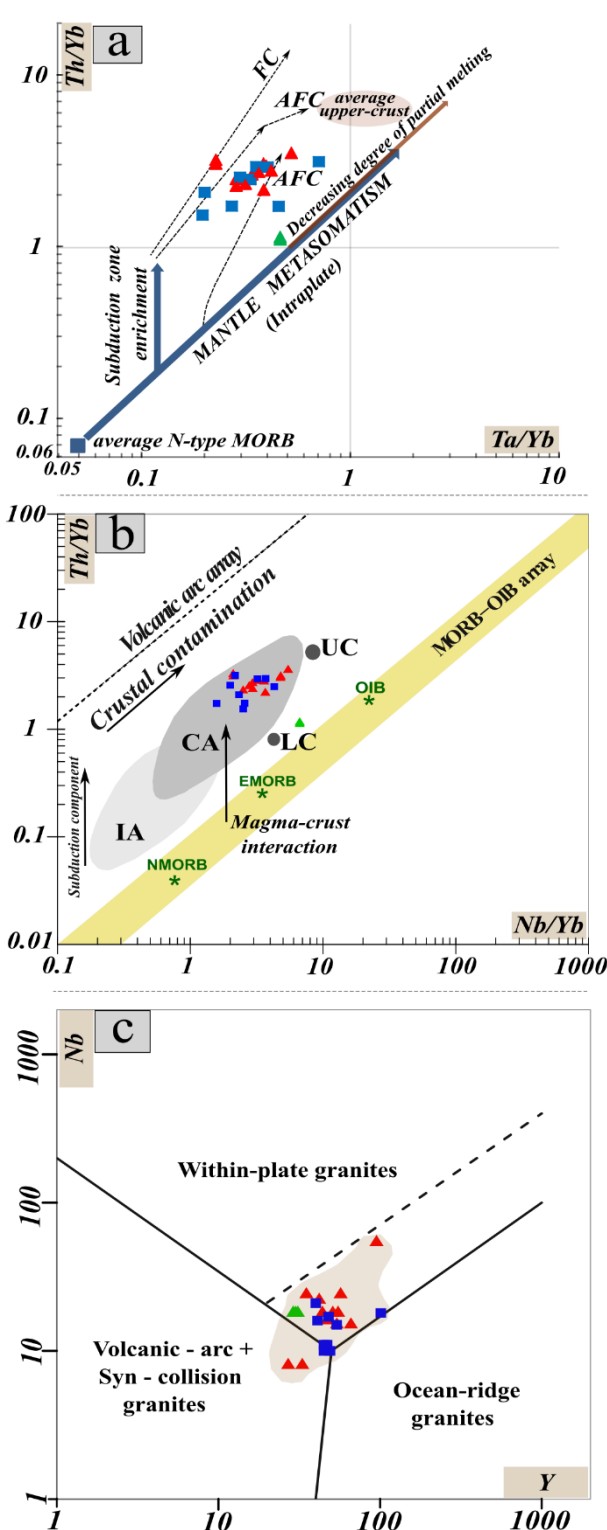




### 5.3 Significance of LA-ICP-MS U-Pb data

#### 5.3.1 U-Pb zircon age record on magmatic rocks of the OG

Reliable new U-Pb zircon ages on two magmatic samples (Zg-106, and Zg-119; Fig. 4), representing the Ouarzazate Group
from the Sirwa massif returned ages of 575 ± 3 Ma and the 564 ± 2 Ma, respectively. These ages are consistent with a lower
Ouarzazate Group affinity, and represent the extension of the Ediacaran magmatism previously described in numerous inliers
of the Anti-Atlas belt (Yajioui et al., 2020; Karaoui et al., 2015; Hefferan et al., 2014, and references therein; Toummite et al.,
2013; Walsh et al., 2012; Gasquet et al., 2005, 2004; Thomas et al., 2002).

Indeed, in the Zgounder Mine Region, similar volcanic activity of this type can be bracketed between 620 Ma to 550 Ma
(Pelleter et al., 2016; Thomas et al., 2002; This study). For the rhyolite (Zg-106); Pelleter et al., (2016) reported mean age of
578 ± 4 Ma from the Zgounder Mine for the rhyolitic dikes and plugs using the 207Pb/206Pb ages. Moreover, in the Sirwa
massif, similar U–Pb ages of 579 ± 7 Ma, 571 ± 8 Ma, 577 ± 6 Ma, 579 ± 7, and 575 ± 8 Ma were proposed for the Tourcht
diorite, the Tikhfist rhyolite (Tiouin Subgroup, Fig. 1C), the Aguins member (Tafrant Subgroup), the Tilsakht granite, and the
Askaoun granodiorite, respectively (Thomas et al., 2002). Given that the 575 ± 3 Ma age of our rhyolite is analytically identical
within errors to the ages of these rocks, thus inferring a rapid sequence of intrusive events that occurred within a span of a few
million years.

For the granite sample (Zg-119); the obtained age of 564 ± 2 Ma is also identical within a margin of error to the ages of the
neighbouring Imourkhssen granite dated respectively at 561 ± 3 Ma, and 562 ± 5 Ma by Toummite et al., (2013), and Thomas
et al., (2002) (Fig. 1C). The Imourkhssen granite on its turn intrudes the lower and older part of the Ouarzazate Group in the
Sirwa massif (Thomas et al., 2002). Additionally, in the Sirwa massif, numerous syn- to late-Ouarzazate Group granites with
a sub-alkaline-calcic composition provided ages of c. 560 Ma, making them the Ouarzazate Group's youngest components.
According to Thomas et al., (2002), these granites belong to the Achkoukchi Complex (Fig. 1C), and grouped into the
Amassine Suite (e.g., Bou-Tazart, Aït Nabdas, and Tikitar granites, and the Tazoult Quartz-porphyry at 559 ± 6 Ma).

Significant inherited Paleoproterozoic zircon population are previously reported for the Ouarzazate Group rocks all over the
Anti-Atlas. For instance, Baidada et al., (2019) reported inherited Paleoproterozoic signature for the Saghro massif granitoids.
Blein et al., 2014b proved the existence of Paleoproterozoic inherited zircon in ignimbrite of the Ouarzazate Group in the
Agadir Melloul area. Finally, Thomas et al., (2002) highlighted Paleoproterozoic inherited zircons in syn-Ouarzazate Group
granites in the Sirwa massif.

Indeed, the Ouarzazate Group's thick successions of volcano-sedimentary series are dominated by acidic and intermediate
high-K calc-alkaline to shoshonitic volcanism (Blein et al., 2014b; Toummite et al., 2013; Walsh et al., 2012; Gasquet et al.,
2008, 2005; Benziane, 2007; Thomas et al., 2004; El Baghdadi et al., 2003). Their depositional environment as pull-apart
basins with strike-slip faulting conditions and sub-vertical movements favored a high variability in thickness (Walsh et al.,





2012, and references therein). Nevertheless, a contemporaneous magmatic activity spanned the 630 to 538 Ma time frame; (Table 1 in Hefferan et al., (2014) for reported precision ages from selected inliers of the Anti-Atlas Mountains). All in all, this activity indicates a prolonged tectono-magmatic event emplaced over multiple pulses over the whole Anti-Atlas belt

(Tuduri et al., 2018). In regard to this, Schulte et al., (2022), considered the Ouarzazate Group in the Central and Eastern Anti-Atlas as the relicts of an Ediacaran silicic large igneous province (SLIP) deposited in a strictly continental environment. These huge volumes of mostly felsic magma were emplaced by multiple pulses and occur as a result of a long-lived magmatic event evolving from high-K calc-alkaline at 575–550 Ma to alkaline affinity at 550–540 Ma (Schulte et al., 2022; Blein et al., 2014b; Gasquet et al., 2008). This magmatism is considered to belong to a continental silicic large igneous province (SLIP), referred

to as the Ouarzazate Silicic Large Igneous Province (OSLIP), and emplaced over multiple periods at ca. 575 Ma, ca. 560 Ma, and ca. 550 Ma (Tuduri et al., 2018; Blein et al., 2014b). Consequently, the last two pulses at the final stage of the Pan-African orogeny (560 to 550 Ma) appear linked to widespread pervasive hydrothermal activity across the whole Anti-Atlas (Tuduri et al., 2018). Regional correlations can be drawn to the extent of the Cadomian orogeny (Linnemann et al., 2014, and references therein). Hence, our samples fall within the age range of 580 to 550 Ma of the Cadomian arc magmatism extended to Iberia

(Chichorro et al., 2022, and references therein).

### 5.3.2 The Saghro Group detrital age: depositional style and material source

Whether or not the Paleoproterozoic basement exists in the Sirwa massif north of the AAMF, and by consequence the northern margin of the WAC is still debated (Ennih and Liégeois, 2008). Outcrops of this Paleoproterozoic basement have been regarded as being present exclusively in the westernmost part of the Anti-Atlas belt (Choubert, 1963). However, new studies based on

TDM ages support the hypothesis that Paleoproterozoic rocks may also be found in both the Central and Eastern segments of the Anti-Atlas (Baidada et al., 2019; Blein et al., 2014; Toummite et al., 2013; Liégeois et al., 2013; Abati et al., 2010; Gasquet et al., 2008; Thomas et al., 2002). Moreover, this hypothesis can be extended to Western Meseta (Pereira et al., 2015, and references therein; Tahiri et al., 2010; Baudin et al., 2003).

During the last decade, the northern boundary of the WAC is considered to be marked by the northernmost outcrops of highly

deformed rocks in the southern part of the Bou-Azzer inlier, particularly the Tazagzaout migmatites and gneisses, based on similarities in lithology and deformation degree with the Zenaga Complex in the Western Anti-Atlas (Leblanc and Lancelot, 1980). However, this view has been changed owing to the new U-Pb precision dating obtained from the Tazagzaout gneisses ($752 \pm 1^{-}_{2}$ Ma; D'Lemos et al., (2006), and the Oumlil granite $741 \pm 9$ Ma; El Hadi et al., (2010)). Furthermore, the positive Nd values of (+ 4.9 to + 6) for the Tazagzaout Complex refer to a juvenile depleted mantle source, not akin to the 2 Ga

Eburnean WAC basement (D'Lemos et al., 2006). Consequently, the presence of a WAC basement beneath the Central Anti-Atlas, especially in the Sirwa massif is not clear. Published geochronological data indicate that both the Saghro and Bou Salda groups were deposited approximately 630 to 610 Ma. This deposition occurred during a period of tectonic convergence of the Cadomian arc upon the WAC Iriri-Tazagzaout Complex (Errami et al., 2021a; Walsh et al., 2012; El Hadi et al., 2010). Yet, the precise location of the suture zone marking this collision remains ambiguous. Current interpretations suggest the suture




likely corresponds to the Anti-Atlas Major Fault (AAMF) (Hefferan et al., 2014), potentially extending north along the South Atlas Fault (SAF) (Ennih and Liégeois, 2001, 2008).

Detrital zircon age from our study provides new insights into the basement beneath the Sirwa massif. Sample (Zg-132) from the Imghi Formation of the Saghro Group returned exclusively Paleoproterozoic ages, with a prominent peak at ca. 2100 Ma (Fig. 4-3). Interestingly, no ages younger than 1600 Ma were found in our sample, nor were any zircons dating to the local

Pan-African magmatic period in the Anti-Atlas identified. These 883 - 640 Ma Pan-African younger zircons, which are common in outcrops of the Sirwa inlier and have been consistently reported in Saghro Group sediments (Letsch et al., 2018; Abati et al., 2010) are absent in our sample. Indeed, this Paleoproterozoic peak goes in line with the inherited signature from the analyzed zircon population of sediments of the Saghro and Bou Salda groups from the Sirwa massif, for which the obtained maximum depositional ages cluster around 620 – 610 Ma (Abati et al., 2010). Arguably, the authors argued in favor of an

exposure of cratonic basement (Paleoproterozoic ?) based on the relatively high proportion of the 610 Ma zircons in regard to Paleoproterozoic zircons in the upper levels of the stratigraphy sequence. Therefore, this suggests that the source of the 610 Ma was eroded favoring the enrichment in Paleoproterozoic ages. Admittedly, the subsequent erosion of the underlying Neoproterozoic rocks (notably synchronous magmatic rocks, and Saghro and Bou Salda groups) might be the source for the prominent 610 Ma peak in the Iberian massif (Chichorro et al., 2022); (Fig. 13A).

The mono-peak at 2.1 Ga can only be accepted to strictly represent Paleoproterozoic. It can be interpreted as representing the material source for the Saghro Group sedimentary units only in the study area (Fig. 13A). Indeed, the returned mono zircon population may reflect insufficient sampling of detrital zircons. However, the number of analyzed zircons (up to 139; see supplementary table 5) is considered adequate to address paleogeographic questions and sediment source areas. In this context, we interpret the Paleoproterozoic mono-peak in our sample to reflect local exposure of Paleoproterozoic basement rocks along

the basin margins of the Saghro Group in the study area (Fig. 13A). Consequently, most sediments were derived exclusively from the erosion of these basement rocks in a possible mono Paleoproterozoic paleo-relief without interaction with Cryogenian sediments.

Overall, this evidence argues that during Ediacaran times, the Paleoproterozoic basement was in fact totally or locally exposed in the Sirwa inlier (Fig. 13A). This interpretation is further corroborated by Nd model ages of Ediacaran magmatism from this

study (see section Sm-Nd), which yield TDM age of Mesoproterozoic values, indicating a mixing/recycling of older Paleoproterozoic crust with juvenile Ediacaran magma. All in all, and based on these findings, the WAC crust extends beneath the Sirwa massif and likely continues northward far beyond the Anti-Atlas belt. Notably, Paleoproterozoic rhyolites have been reported in the Meseta Block, north of the Anti-Atlas, with a given age of 2050.6 ± 3 Ma (Pereira et al., 2015). These findings do support the extension of the WAC Paleoproterozoic crust northward beyond the AAMF, which was previously considered

the northern margin of the craton.







**Figure 13: Reconstruction of the geotectonic setting of Saghro Group and Ouarzazate Group during Ediacaran times. (resembled and modified after El Kabouri et al., 2025; Chichorro et al., 2022; Errami et al., 2021a; Abati et al., 2010; Thomas et al., 2002). (A): Saghro Group deposition in a back-arc basin (This study; Ouguir et al., 1996), implying local exposure of the 2.1 Ga Paleoproterozoic**
**crust in the Zgounder Mine Region (This study). (B): Pan-African transpressional regime and collision in the Anti-Atlas, resulting in Saghro Group folding, injection of the 600 Ma calc-alkaline magma (Errami et al., 2021a), and proto molasses basin formation (Thomas et al., 2002). (C): General uplift in the Anti-Atlas. (D): Orogenic collapse favoring the thick Ouarzazate Group sedimentation (e.g. post-tectonic molasse deposition and calc-alkaline magmatism, with coeval acidic rocks).**

## 6 Conclusions

In the light of the above, the following conclusions can be drawn:

   i. Based on geochemical proxies (major and trace elements), the Saghro Group rocks exhibit both active continental margins and within-plate characteristics. A dual signature characteristic of magmas generated during back-arc extension. Therefore, the recognized subduction-related signature on our samples is acquired through mantle metasomatism by sediment-derived components during arc activity, while the within-plate signature emerges later
during back-arc extension processes.

   ii. The Saghro Group rocks were emplaced at the early stages of back-arc basin opening, testified by their calc-alkaline affinity, geochemical characteristics and regional correlations. They were derived from contaminated mantle source with an old continental crust. Further, their εNd (at 620 Ma) = + 3.2 to + 4.5, and TDM ages of 1431 – 1197 Ma reflects a mixed origin, combining mostly mantle-derived magma with limited proportion of older crustal material.
Thus admitting the existence of an old Paleoproterozoic to even Mesoproterozoic crust under the Saghro Group.

   iii. For the Ouarzazate Group rocks, the evolution in chemical compositions for the suite of high-K calc-alkaline to shoshonitic rocks is controlled in the most part by fractional crystallization and crustal contamination. They were deposited in a post-collisional setting, and their primary magma was sourced from an enriched continental lithospheric mantle, previously underwent metasomatism by fluids from former Neoproterozoic subduction. Further, their εNd (at
570 Ma) = - 0.9 to + 1.1, and TDM ages of 1526 to 1252 Ma refer to a Mesoproterozoic (and Paleoproterozoic) affinity, implying recycling of Paleoproterozoic material during Ediacaran times.

   iv. LA-ICP-MS U-Pb zircon ages on magmatic rocks evince an expression of a post-collisional syn-orogenic magmatism (WACadomian arc), during the emplacement of its first and second pulses at ca. 575 to 560 Ma. This period of magmatism is concordant with a Silicic Large Igneous Province (SLIP) recognized all over the Anti-Atlas.

**Code, data, or code and data availability**

Data will be made available upon request.





**Supplement link**

**Author contributions**

**Abdelhay Ben-Tami:** Field work, Sampling and preparation, Thin-sections preparation, Investigation, Data visualization, Data curation, Formal analysis, Writing of original draft, Finalization. **Said Belkacim:** Supervision, Resources, Writing - review and editing. **Jamal El Kabouri:** Data visualization, Writing of original draft. Review and editing. **Joshua H.F.L. Davies and Morgann G. Perrot:** LA-ICP-MS U-Pb data and methodology, writing of original draft, review and editing. **Mariam Ferraq**: Data curation and visualization. **Mohamed Bouabdellah:** review and editing. **Bouchra Baidada, Mohamed Bhilisse** and **Mohamed Assalmi:** Resources, accommodation, field trips, hosting. **David Lalonde:** Resources, Financing, Collaboration. Review, and validation.

**Competing interests**

The authors declare that they have no conflict of interest. Corresponding author (Abdelhay Ben-Tami), discloses an employment relationship with the Zgounder Millenium Silver Mining Company (ZMSM).

**Disclaimer**

Copernicus Publications remains neutral with regard to jurisdictional claims made in the text, published maps, institutional affiliations, or any other geographical representation in this paper. While Copernicus Publications makes every effort to include appropriate place names, the final responsibility lies with the authors. Views expressed in the text are those of the authors and do not necessarily reflect the views of the publisher.

**Acknowledgements**

This research represents a component of the corresponding author's thesis project. The authors express their sincere gratitude to the **Zgounder Millenium Silver Mining Company (ZMSM)**, the Moroccan subsidiary of **Aya Gold and Silver (AYA)**, for their essential financial support for this study, as well as for providing accommodation and logistical assistance. Special appreciation is extended to the Exploration geology team at the Zgounder Mine for their valuable facilitation and support during field operations. Official permission to publish this work was granted by **David Lalonde, P. Geo / VP Exploration / Aya Gold and Silver Inc.** Finally, the authors wish to thank the anonymous referees for their constructive feedback and suggestions, which significantly enhanced the quality of this manuscript.



**Financial support**

Financial support for the Whole-rock analysis, Sm-Nd isotopes, U-Pb on zircon, fieldwork and accommodation was guaranteed by the **Zgounder Millenium Silver Mining company (ZMSM)**; the Moroccan branch of **Aya Gold and Silver Inc**.

**Review statement**

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
