# Peer review of "Petrogenesis and geodynamic implications of Ediacaran rocks from the Sirwa massif (Central Anti-Atlas); insights from U-Pb geochronology, whole-rock geochemistry, and Sm-Nd isotopes"

_EGUsphere, 2025_

## Referee Comment (RC1)

Dear Editor,

Manuscript "Petrogenesis and geodynamic implications of Ediacaran rocks from the Sirwa massif (Central Anti-Atlas); insights from U-Pb geochronology, whole-rock geochemistry, and Sm-Nd isotopes" by Abdelhay Ben-Tami and co-workers represents a potentially interesting contribution to the ongoing discussion concerned with the nature and petrogenesis of Neoproterozoic (Pan-African) magmatic and sedimentary rocks in the Anti-Atlas Belt (Morocco), and their geodynamic significance. The text brings new, precise whole-rock geochemical dataset, including some Nd isotopic compositions, in situ LA ICP MS U–Pb ages of zircon. However, the presentation, interpretation and discussion of these data could be improved significantly.

The text is in places confusing, wordy, and/or difficult to follow. Also, the petrogenetic models should be better introduced and justified. The same is valid also for the geodynamic setting. In fact, after reading the manuscript carefully, I am still not sure what the preferred scenario is.

Hereafter I give some, usually more general comments; for the more concrete remarks/edits, please refer to the attached annotated PDF document.

To sum up, I cannot recommend the publication of the reviewed manuscript in the present form, nad requires revision. I trust that the authors will be able to revise the manuscript fundamentally, incorporating most of the changes and addressing my criticism, so that it could be published and find its interested readers.

Prof. Vojtěch Janoušek
Czech Geological Survey/Charles University, Prague

**General comments to the style, language and grammar of the manuscript**

- The Results and Discussion parts should be strictly separated.

- The manuscript is written in English with numerous mistakes. The language requires revision by native speaker. The text tends to wordy and repetitive, and in places does not flow linearly. It can be certainly condensed substantially without a loss of its informative value. Currently it is extensively long. I urge the senior co-authors (and native speakers among them) to take care of these issues.

- Reduce, if possible, the plethora of local names. All of these remaining need to be shown on some map.

- Consider capitalization of the whole formal names, including words such as Orogeny, Group, Pluton or Complex.

- All abbreviations are to be explained just once, at the first occurrence. Except in figure captions, where I would always explain all of those present.

- Past tense should be used always when referring to events in geological past or previous publications.

- Hypotheses should be expressed by some element of uncertainty, e.g. by conditionals.

- In English text (e.g., Table 2), one should use exclusively decimal points, and not commas.

- All the measured/calculated values should be rounded up to their precision.

- All Figures (and their parts), Tables and electronic supplementary materials (ESM) should be quoted in the text, and exactly in the correct sequence.

- Define how you calculate some geochemical parameters. For Eu anomalies, use the standard Eu/Eu* notation. Add all these extra parameters to the data table.

**Graphics**

- The quality and readability of all graphs should be improved. The isotopic plots are terrible.

- Plotting symbols/colours should be chosen carefully and kept the same for all plots.

- In geochemical plots, label the samples specifically mentioned in the text.

- Add references as appropriate, including those referring to the plotted geochemical reservoirs and trends. Format those on plots according to the Journal in house rules.

- Avoid using underscore, _ as a replacement for space.

- In some figures (most notably Fig. 1), text seems to be interlaced, and is very difficult to read. But maybe it is just some compatibility issue.

**Typography**

- More attention should be paid to correct, and consistent, typography, placing spaces where necessary, correctly using hyphens and en dashes, etc. For example, the ranges should be indicated by en-dashes, without spaces around. Minus is en-dash, with no space following.

- Special care should be paid to correctly marking super- and subscripts, e.g. in names of major-element oxides.

- There should be always a space between a number and unit, or a percent symbol. There is no space before punctuation marks such as .:;.

- There should be no space between a sign and the number, e.g. +5.5 or –2.7.

**References**

- The authors should adhere strictly to the journal's style as given in the Instructions to authors.

- What is the logic behind ordering of multiple references in the text? I would say that they are currently completely erratic. Order them consistently; chronologically, or alphabetically, depending on your preference.

- The same applies for Bibliography. See https://www.solid-earth.net/submission.html#references. Check that if there is more than one paper in the same year for a first author (independent of the rest of the team), a letter (a, b, c) is added to the year both in the in-text citation as well as in the reference list.

- The bibliography is rather accurate; there is only a handful of references missing in the list, see the annotated PDF.

**Abstract**

[23] Instead of sample names, specify the lithologies dated.

[28] Formed from a dominantly juvenile, mantle-derived source – confusing. Do you mean direct contribution of mantle-derived basaltic melts or remelting of metabasic crust?

**Introduction**

[39–43] Complex sentence, revise.

[47–48] Repetitive. It was said above, combine.

[58] Sirwa or Siroua as on the map?

[60] Mafic and intermediate units? Please be more specific.

**Geological setting**

[73] Distinguish between orogeny (the orogenic process) and orogen (its product).

[114–148] Are all these details and local names indeed necessary?

[128] Provide errors of age determinations, and dated material/method, whenever possible.

[128] Slightly younger? Probably identical within the error.

[Table 1] Should be moved to electronic supplement. What is the logic behind ordering the individual samples?

**U–Pb zircon geochronology – methodology**

[154]    Show sample Zg-119 on the map.

[159]    ± 2 sigma error?

**Petrography**

[175]    I would rather call this texture ophitic(?)

[189]    pyroxene? Could you please be more concrete, or at least to write "clinopyroxene"?

[189]    List the minerals in the order of decreasing modal abundance.

**U–Pb zircon geochronology**

[220]    Moderate in size? Please be more concrete.
The CL images are too tiny to see any detail.

[235–236]   Both? What do you refer to?

[238]    Poor style. The age of the sandstone Zg 132 has not been introduced yet.

[243–252]   Perhaps it would be interesting to discuss the youngest zircons constraining the maximum age of sedimentation.

**Whole-rock geochemistry**

[265]    But, if I understand it right, a lot of them are altered.

[274] In general, label the samples specifically mentioned in the text on the geochemical plots.

[280 and elsewhere]        The Eu anomalies should be expressed in consistent way. δEu, Eu/Eu*)N Eu/Eu* are all used. Eu/Eu* is the standard.

[284] Be specific: what is the "early gabbro sample from Sirwa"?

[296] What is the unit of $Na_2O + K_2O$? Is their ratio calculated by weight?

[297–298] These rock groups are not distinguished in the plot.

[299] Some nomenclature diagram is to be shown to name the rocks discussed in this text.

[309] This is a hypothesis that should not appear in the Results, but in Discussion. K-feldspar fractionation would have the same effect.

[316–320] Very confusing. First of all, such a discussion is out of scope of the Results section. There is a lot of local information we are not familiar with and these analyses are not plotted here.

ESM 4

- In text and in the relevant ESM table: round off all the data to match the precision of each of the oxides/elements (some are precise to three decimal places, others to none, I suspect). Even add trailing zeroes whenever needed to indicate the real precision.

- Mg numbers are calculated wrongly! The calculation should be based on molar and not weight percentages.

**Sm–Nd isotopes**

- The Analytical techniques (ESM 2) lack the necessary details of Nd isotopic data recalculation, including the relevant references for decay constant, CHUR composition and model ages computations.

Table 2

- Sample names should come into the first column.

- Add a column with age (Ma) used for correction each of the analyses.

- Format all super- and subscripts as appropriate.

- Show initial Nd ratios, but not the present-day epsilon values.

- The column with initial ratios should be labelled $\varepsilon_i^{Nd}$ or similar. Some analyses were not recalculated to 570 Ma.

- Model ages should be given in Ga and rounded to two decimal places.

- What is the difference between *TDM (Goldstein) Ma* and *TDM Ga*? Are these single- or two-stage models?
  Give all these details of model age calculations in the Analytical techniques. As you know, single-stage ones are more appropriate for mantle-derived, mafic rocks, two-stage model ages (Liew and Hofmann 1988) for crustally-derived, felsic rocks.

- The initial ratios of Nd isotopes and epsilon values should have a consistent – and simple – symbology. Ideally, they should be labelled by subscripts '570' or '620', directly indicating the age used for their correction. E.g. $\varepsilon_{570}^{Nd}$.

[338] There are just two values for Saghro rocks, it is thus misleading to describe them as an interval.

[341–343] This is misconception! Intermediate positive epsilon values *per se* may be equally well explained by derivation from less depleted mantle domains, or remelting of fairly juvenile metabasic crust. Additional evidence is needed.

[345–346] This, and some other parts of this section, should move to Discussion.

[351–352] This again should be moved to the Discussion, and additional petrological and geochemical evidence should be considered, also from literature.
What do you mean, a mantle source contaminated by older crustal component? On which evidence? Why not, for instance, AFC or assimilation of crustal material during ascent of doleritic magmas? What compositions of mantle and crust do you envisage? Was the mantle close to canonical depleted mantle, or CHUR-like? Do you really assume that the felsic magmas were generated from mantle-sourced magmas? Or did they come from remelting of a pre-existing crust?

[352–353] If the dolerite came from CHUR-like mantle, the DM model age is just meaningless.

[352–359] Again, this belongs to Discussion.

[358–359] Last sentence seems an overinterpretation of your data.

Figure 8:

- Plotting symbols/colours should be the same like in other plots. Here are even used the same symbols in panels a–b vs. c–d for different things!

- It is graphically very poor, having been apparently exported directly from Excel. It lacks any formatting – superscripts, epsilon letters etc. Needs to be redrawn.

- I suggest a better representation – in the form of two plots: (1) a Nd growth diagram (age vs. epsilon Nd, it can also incorporate the model ages and (2) some independent geochemical parameter (e.g., $SiO_2$), vs. initial epsilon Nd values. See below, BTW graphs were generated by our software GCDkit (Janoušek et al. 2006) and obviously would need to be supplemented by literature data.

[Figure]

**Discussion – Petrogenesis**

This chapter is very confusing and scientific argumentation is not sufficient or misleading. It needs to be rewritten completely.

[374–380] There are no such plots shown, neither in the text, nor in the ESM 4. So, the hypothesis cannot be tested. Add these plots, justify your model and do not forget to consider alternatives, such as partial melting. Perhaps better would be diagrams against Mg#, rather than Zr.

[380–385] Mg# are calculated wrongly, so this paragraph needs to be rewritten. What is the evidence for this mixing?

[386] I do not know this "contamination-sensitive" diagram. Contamination by what material? How is it supposed to work? Give a reference. Note that in most magmatic rocks, La and Ce will be strongly corelated and behaving very similarly during fractional crystallization or partial melting. I guess this projection is of very little value in distinguishing closed-system differentiation from continental crustal contamination and I would drop it.

Using Nd isotopes for this purpose is definitely a much more powerful approach.

[389–392] But not like this! The "mixing model" in ESM 4 is calculated without taking the contrasting Nd concentrations (ppm) in the both end-members. The correct approach is (Janoušek et al. 2016):

The mixing equation for two end-members, *1* and *2*, can be seen as a mean of isotopic ratios ($I_1$ and $I_2$), weighted by their respective mass-fractions in the mixture. Marking the mass fraction of the end-member *1* as $f_1$, (with $f_1 + f_2 = 1$) and respective concentrations $C_1$, $C_2$ and $C_M$ (Faure 1986):

$$I_M = I_1\left(\frac{C_1}{C_M}\right)f_1 + I_2\left(\frac{C_2}{C_M}\right)(1 - f_1) \qquad (0.1)$$

Where:

$$C_M = f_1 C_1 + (1 - f_1)C_2 \qquad (0.2)$$

[398–340] I cannot see this, rephrase. Perhaps this is not the best projection, either.

[402–403] This diagram is designed solely to judge the nature of crustal protoliths melted. It does not make sense for mantle-derived rocks and also does not show any effects of

fractional crystallization and contamination of primary magmas. Revise.

[404] Enriched in what sense? Enriched mantle? The whole sentence is a bit daring and premature. Neodymium isotopes need to be assessed first.

Fig. 9

- I would omit literature data, as only several points are plotted on a single panel.

- Instead of panels a, b I would rather plot Nb/Yb vs. Th/Yb and Nb/Yb vs. TiO$_2$/Yb (Pearce 2008) – the first you have as a current Fig. 12b.

- Panel b: normalized by what (reference).

- Panel c: IAT is not shown.

- All abbreviations need to be explained. Give references for compositions of each of the reservoirs plotted.

[411] Why there are not discussed effects of alteration like in the case of SG above?

[414] Again, these diagrams are not plotted.

[421] Calcic phase? Feldspars.

[423–425] High LREE contents can be also due to direct derivation from crustal sources and would be further modified by fractional crystallization.

[424–425] Speculations.

[427] The "trend" is too scattered to reveal anything. The plutonic rocks are not distinguished on the plot.

[431–432] Hydrous metasomatism? What do you mean? Why should be P anomalies linked to Nb and Ti depletions?

[442] Or Nb and Ti anomalies can reflect subduction setting.

[443–444] I cannot understand the bit starting from "as supported by epsilon Nd values…"

Fig. 10

- Not only OG, but also SG rocks. Add legend.

- Technically speaking, Harker plots are binary diagrams of SiO$_2$ vs. major elements, not traces.

- Nickel will not show anything else than Cr, and determinations of the former are rounded to tens ppm, so rather imprecise. Omit. Are the rocks fresh enough so that Rb and Ba can be used?

- How were the BA, AFC and FC trends obtained? Reference? Assimilation of what? How much? Fractional crystallization of what minerals and in which proportions? What was the degree of crystallization? What Kd values were used (references)?

- Panel b – for intermediate and felsic rocks, the La concentration will be controlled, to a large extent, by accessories, such as allanite or monazite.

- Panel c – why Y/Nb of 1.2? Would not be Nd isotopes much better for this purpose?

[445] What is "enriched continental crust"?

[448] Moderate epsilon Nd values? Not clear, be more specific. Everything depends on local mantle composition, it could have been CHUR-like easily.

[450] What is "sediment zone enrichment"?

[448–452] I am completely lost. What is your preferred scenario? Genesis from earlier (oceanic) subduction-modified mantle or contamination (AFC) of E-MORB-like melts by continental crust?

Fig. 11

- Explain all abbreviations and give the references for compositions of various average mantle and crustal reservoirs.

- Panel c: missing is explanation of the trends for various processes. How were they constructed (see also my previous comment)? BABB symbols look just terrible.

Fig. 12

- Explain all abbreviations and give the references for compositions of various average mantle and crustal reservoirs.

- What is the difference between the Ta/Yb vs. Th/Yb and Nb/Yb vs. Th/Yb projections? Should be giving moreless the same information….

- Panel a: missing is reference, explanation of the trends. How were they constructed?

**Discussion – Geodynamic implications**

Again, this section requires a thorough revision.

[466] Show it! This is not acceptable.

[469] Or simply rifting. I would suggest some other diagrams that could help you resolving the geodynamic setting, namely those of Wood (1980), Cabanis and Lecolle (1989), Pearce et al. (2021) or Shervais (2022). Surprisingly, the basic rocks of the SG and OG look like continental flood basalts in the Ti/V vs. Th/Nb plot of Shervais (2022). I do not know much about local geology, but is any role of plume ruled out?

[Figure]

[Figure]

[Figure]

TiO$_2$/Yb – Th/Nb (Pearce et al. 2021)      Ti/V – Th/Nb (Shervais 2022)

[472] Why is low U/Th ratio indicative of active margin signature?

[473–474] Normalized to what? To my eyes, the SG nicely follow the NMORB–OIB array. CAB should have higher ThN.

[476–479] References missing.

[501] A fragment, a sentence should have a verb. Rephrase. Fig. 11a shows Y vs. Zr, not K, Rb, Ba etc. Plus, these are extremely mobile elements, could not they be compromised by alteration?

[508] Be specific, how?

[510] And what was the cause of such post-collisional event?

[515] evolution?

**Discussion – U–Pb ages**

[518] Please specify the sample lithologies, this is more important than the sample numbers.

[520] Extension of the magmatism? Rephrase.

[532] How about the dating by Ferraq et al. (2024)?

[535] What is sub-alkaline-calcic?

[538] Did you observe any inheritance?

[549] What is "a prolonged tectono-magmatic event emplaced over multiple pulses over the whole Anti-Atlas belt"?

[551] "SLIP deposited in a strictly continental environment"? I cannot follow this.

[551–556] Tedious and repetitive. Condense.

[558–560] Rephrase and expand the part dealing with regional correlation of the Cadomian arc magmatism and how does it relate to the inferred geodynamic setting of the studied rock units.

[590–593] I cannot follow this argument. Clarify.

[595] "The mono-peak…" does not make any sense to me.

[601] Cryogenian should be older than 635 Ma. These sediments are Ediacaran, or younger.

[610] Reference missing here.

Fig. 12

Explain all the abbreviations.

**Conclusions**

[627–628] What is the evidence for contamination of the mantle source by continental crust? How does it go with the presumed oceanic subduction context?

[636] How about your sediments?

[637–638] "post-collisional syn-orogenic magmatism (WACadomian arc)"? I am again lost. Regardless, how can you infer this from U–Pb ages only?

**References used**

Cabanis B, Lecolle M (1989) Le diagramme La/10–Y/15–Nb/8: un outil pour la discrimination des séries volcaniques et la mise en évidence des processus de mélange et/ou de contamination crustale (The La/10-Y/15-Nb/8 diagram: a tool for discrimination volcanic series and evidencing continental crust magmatic mixtures and/or contamination). C R Acad Sci Paris, Série II 309:2023–2029

Faure G (1986) Principles of Isotope Geology. 2 edn. John Wiley & Sons, Chichester

Ferraq M, Belkacim S, Cheng L-Z, Davies JHFL, Perrot MG, Ben-Tami A, Bouabdellah M (2024) New geochemical and geochronological constraints on the genesis of the Imourkhssen Cu±Mo±Au±Ag porphyry deposit (Ouzellagh-Siroua Salient, Anti-Atlas, Morocco): geodynamic and metallogenic implications. Minerals 14:832

Janoušek V, Farrow CM, Erban V (2006) Interpretation of whole-rock geochemical data in igneous geochemistry: introducing Geochemical Data Toolkit (GCDkit). J Petrol 47:1255–1259

Janoušek V, Moyen JF, Martin H, Erban V, Farrow C (2016) Geochemical Modelling of Igneous Processes – Principles and Recipes in R Language. Bringing the Power of R to a Geochemical Community. Springer, Berlin

Liew TC, Hofmann AW (1988) Precambrian crustal components, plutonic associations, plate environment of the Hercynian Fold Belt of Central Europe: indications from a Nd and Sr isotopic study. Contrib Mineral Petrol 98:129–138

Pearce JA (2008) Geochemical fingerprinting of oceanic basalts with applications to ophiolite classification and the search for Archean oceanic crust. Lithos 100:14–48

Pearce JA, Ernst RE, Peate DW, Rogers C (2021) LIP printing: use of immobile element proxies to characterize Large Igneous Provinces in the geologic record. Lithos 392–393:106068

Shervais JW (2022) The petrogenesis of modern and ophiolitic lavas reconsidered: Ti-V and Nb-Th. Geosci Front 13:101319

[revised manuscript text omitted]